# Rationally designed synthetic protein hydrogels with predictable mechanical properties

Junhua Wu[1,2], Pengfei Li[1], Chenling Dong[3,4], Heting Jiang[1], Bin Xue[1], Xiang Gao[1], Meng Qin[1], Wei Wang[1], Bin Chen[3,4] & Yi Cao [1]

Designing synthetic protein hydrogels with tailored mechanical properties similar to naturally occurring tissues is an eternal pursuit in tissue engineering and stem cell and cancer research. However, it remains challenging to correlate the mechanical properties of protein hydrogels with the nanomechanics of individual building blocks. Here we use single-molecule force spectroscopy, protein engineering and theoretical modeling to prove that the mechanical properties of protein hydrogels are predictable based on the mechanical hierarchy of the cross-linkers and the load-bearing modules at the molecular level. These findings provide a framework for rationally designing protein hydrogels with independently tunable elasticity, extensibility, toughness and self-healing. Using this principle, we demonstrate the engineering of self-healable muscle-mimicking hydrogels that can significantly dissipate energy through protein unfolding. We expect that this principle can be generalized for the construction of protein hydrogels with customized mechanical properties for biomedical applications.

[1] Collaborative Innovation Center of Advanced Microstructures, National Laboratory of Solid State Microstructure, Department of Physics, Nanjing University, Nanjing 210093, China. [2] Jiangsu Key Laboratory of Molecular Medicine, Medical School, Nanjing University, Nanjing, Jiangsu 210093, China. [3] Department of Engineering Mechanics, Zhejiang University, Hangzhou 310027, China. [4] Key Laboratory of Soft Machines and Smart Devices of Zhejiang Province, Hangzhou 310027, China. Junhua Wu, Pengfei Li and Chenling Dong contributed equally to this work. Correspondence and requests for materials should be addressed to W.W. (email: wangwei@nju.edu.cn) or to B.C. (email: chenb6@zju.edu.cn) or to Y.C. (email: caoyi@nju.edu.cn)

Hydrogels have been widely explored as an artificial extracellular matrix (ECM) material in tissue engineering, regenerative medicine, stem cell and cancer research, cell therapy, and immunomodulation to provide specific three-dimensional (3D) environments that are essential to the culture of and interaction with encapsulated cells[1–7]. The mechanical properties of hydrogels are important for these applications as they provide not only mechanical support but also biophysical stimuli, thereby regulating cell proliferation, spread and differentiation[8–17]. However, the methods that can be used to control the mechanical properties of hydrogels are very limited and are mainly focused only on the elasticity of hydrogels[12,13,16,17]. Other mechanical properties, such as extensibility[18], toughness[19], strain softening/stiffening[20], and mechanical memory[21], which are important properties of native ECMs, are less well characterized. It is highly desirable to engineer hydrogels with controllable and specific mechanical properties.

The mechanical properties of native ECMs are finely controlled due to the presence of folded protein domains[22–25], which tend to be both highly extensible and extremely tough. Conversely, most synthetic hydrogels are composed of unstructured proteins, and therefore do not typically exhibit these properties[26–29]. A few biomimetic designs using folded globular proteins as building blocks have yielded highly elastic and tough protein hydrogels[30], paving the way to design tough and elastic hydrogels[30,31]. Notably, with the development and application of single-molecule force spectroscopy techniques, our understanding of the mechanical properties of individual proteins has significantly advanced in the past few decades[32–37]. However, using these protein-building blocks to rationally design protein hydrogels with independently controllable key mechanical features such as Young's modulus, toughness, and extensibility, remains a fundamental challenge. To truly achieve this goal, it is critical to understand the link between the mechanical properties of individual proteins and the resulting hydrogels.

Theoretical studies suggested that the unfolding of protein structures can lead to large energy dissipation, thus granting the resulting synthetic protein hydrogels and fibers high elasticity and toughness[38–42]. However, in experimental work pioneered by Li and coworkers, the authors argued that on deformation, the forces experienced by protein domains in the hydrogels were as low as only a few piconewtons (pN) even at a strain of 100–200%[30]. Therefore, most folded proteins have sufficient mechanical stability that they cannot be unfolded by hydrogel deformations. Although it was possible to engineer hydrogels mimicking the high toughness and elasticity of muscle by including folded protein domains[31], it remains unknown whether the massive energy dissipation in such hydrogels is mainly associated with the unfolding of the folded protein structures. On the contrary, some hydrogels made of the same folded protein domains and other mechanically strong proteins were mechanically labile[31,43]. Obviously, there is a big gap between the mechanical properties of individual protein building blocks and that of the resulting protein hydrogels.

When force is applied to a hydrogel, the polymer network of the hydrogel will respond accordingly and deform to translate the force down to the molecular level. There are two types of mechanical elements that define the mechanical response of the hydrogel at the molecular level: the cross-linkers (CLs) and the load-bearing modules (LBMs)[44–46]. CLs are responsible for the force transduction among different protein chains and LBMs determine the mechanical response of individual protein chains. For hydrogels made of unstructured proteins or polymers, the deformation of LBMs is mainly entropic, and hydrogel fracture is due to the breaking of CLs[45]. However, if the hydrogel is cross-linked by specific protein ligand–receptor interactions and the

LBMs contain globular proteins, the mechanical response of the hydrogel network is governed by the unfolding/refolding of the LBM in addition to reversible bond rupture and reformation of the CLs, giving rise to a complex mechanical response[46]. We hypothesize that the mechanical properties of the synthetic protein hydrogels are predictable based on the mechanical properties of both the CLs and the LBMs and, more importantly, their different combinations. Accordingly, the mechanical response of the fold protein domains in hydrogels are dependent on the mechanical hierarchy of the CLs and LBMs rather than their intrinsic mechanical properties alone.

In this work, combining single-molecule force spectroscopy, protein engineering and theoretical modeling, we show that synthetic protein hydrogels with predictable mechanical properties can be rationally designed using CLs and LBMs with known mechanical properties. Using single-molecule force spectroscopy, we firstly quantify the mechanical properties of individual CLs and LBMs, showing that the hydrogels exhibit the expected mechanical properties on a molecular level. Then, we prove that the mechanical properties of the resulting protein hydrogels at the bulk level can be precisely predicted based on the mechanical hierarchy of the CLs and LBMs measured at the single-molecule level. We further provide a proof-of-principle demonstration of this design principle by engineering self-healable, strong and tough muscle-mimicking hydrogels. We anticipate that the design principle we present here can be extended to the rational design of hydrogels with complex mechanical response for various applications.

## Results

**Design strategy**. The design concept is schematically shown in Fig. 1a. We designed three types of hydrogels (Gels 1–3) predicted to have different mechanical responses (Table 1). Gels 1–3 are made of two components: a multivalent crosslinker (MCL) and an ABA-type block protein (ABA) (Fig. 1a). The MCL comprises four-armed polyethylene glycol (PEG) molecule fused with a Kir2.3 C-terminal tail peptide (denoted as Kir), which can specifically bind with the ABA protein composed of Tax-interacting protein-1 (TIP-1) domains on both ends and the LBM in the center (TIP-1:Kir complex, Fig. 1b). We chose the TIP-1 protein and Kir peptide complex as the CL because they have high binding affinity at the sub-micromolar range[47] to warrant successful hydrogel construction[48]. The LBMs in the ABA-type block proteins are chosen to have distinct mechanical properties (Table 1). In Gel-1, the unfolding force ($F_u$) of the LBM is significantly higher than the break force ($F_b$) of the CL, which is expected to result in brittle hydrogels with high Young's modulus but low extensibility. In Gel-2, $F_u$ is close to zero and much lower than $F_b$, which is expected to lead to hydrogels with high extensibility but low Young's modulus and low toughness. In Gel-3, $F_u$ is greater than zero but still lower than $F_b$ and the hydrogel is expected to be rigid and ductile and possess high toughness.

Note that if the MCL is changed from the four-armed PEG to an eight-armed PEG or a linear multivalent polymer, mechanical properties of the hydrogels are expected to change accordingly because they change the force distributions on LBMs and CLs within the protein network. However, because LBMs and CLs are connected in series in our experimental design, they are always subjected to the same loading force irrespective to the architectures of the MCLs. As such, it is expected that the mechanical trends for different hydrogels can be similar with different MCLs.

**Mechanics of hydrogel building blocks on a molecular level**. We first used atomic force microscopy (AFM) to characterize the

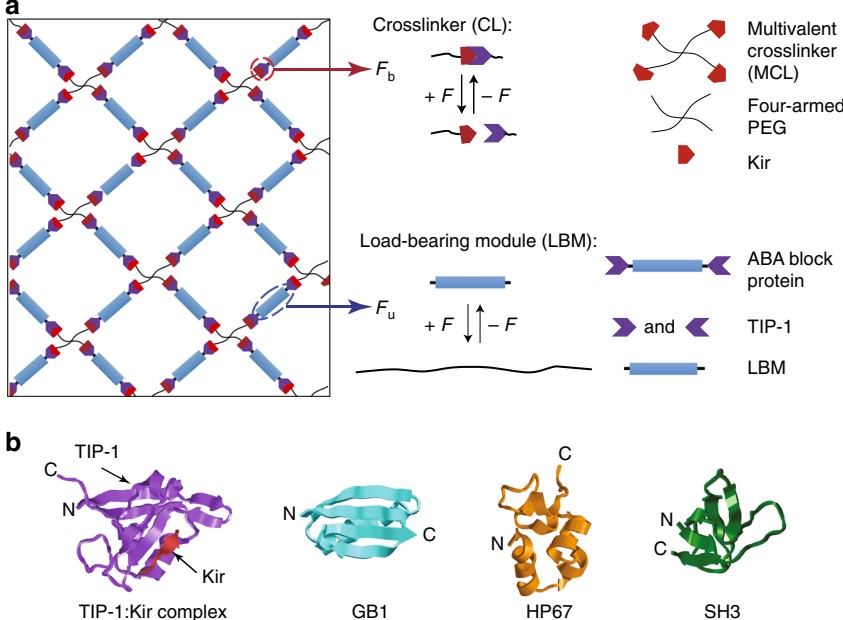

**Fig. 1** Schematic of the hydrogel network design. **a** The hydrogel network is formed from two basic mechanical responsive elements: crosslinkers (CLs) and load-bearing modules (LBMs). The binding/unbinding kinetics of the CL and the folding/unfolding of the LBM are regulated by force. **b** The structure of the protein domains used in the hydrogel design. TIP-1:Kir complex (PDB: 3DJ1), GB1 (PDB: 3GB1), HP67 (PDB: 2RJY), and SH3 (PDB: 1PRL). The structures are generated using Rastop

**Table 1 Designing the mechanical properties of hydrogels at the molecular level and the expected bulk mechanical properties[a]**

| Hydrogel | Cross linker (CL) | Load-bearing module (LBM) | Designed mechanical hierarchies at the molecular level | Expected mechanical properties |
|---|---|---|---|---|
| Gel-1 | TIP-1: Kir $F_b$ (42 or 52 pN) | (GB1)$_8$ $F_u$ (180 pN) | $F_b < F_u$ | Rigid, brittle, self-healing |
| Gel-2 | | (GB1-HP67)$_4$ $F_u$ (<10 pN) | $0 \approx F_u < F_b$ | Soft, extensible, self-healing |
| Gel-3 | | (SH3)$_8$ $F_u$ (25 pN) | $0 < F_u < F_b$ | Rigid, extensible, tough, self-healing |
| Gel-4 | Xmod-Doc: Coh $F_b$ (300 or 600 pN) | (GB1)$_8$ $F_u$ (180 pN) | $0 < F_u < F_b$ | Strong, tough, elastic, function under load, self-healing |

$F_b$ break force of the CL, $F_u$ unfolding force of the LBM
[a]The forces in the brackets were measured at a pulling speed of 400 nm s$^{-1}$

mechanical properties of the CL made of TIP-1:Kir complex. The Kir peptide was linked to an AFM cantilever tip via its N-terminus through a PEG linker. The TIP-1 protein was fused with a Snap domain at its N- or C-terminus to covalently bind it to a glass substrate (based on the well-established Snap-tag protocol Fig. 2a). Pulling apart the TIP-1:Kir complex yielded a single detachment peak in the force-extension curve at an extension corresponding to the contour length of the PEG linker (~50 nm, Fig. 2b, c). When pulling from either the N- or the C-terminus of TIP-1, the rupture forces for the TIP-1:Kir complexes were ~42 and ~52 pN, respectively (Fig. 2d, e). Such a pulling-direction-dependent mechanical strength is a unique feature of force-induced unbinding and has been found in other protein complexes[49]. Detailed dynamic force spectra of the mechanical unbinding of the TIP-1:Kir complexes are illustrated in Supplementary Fig. 1a.

We then tested the behavior of the three load-bearing domains used in the gels that were predicted to have distinct mechanical characteristics. In Gel-1, we used (GB1)$_8$ domains[34] as the LBM (Fig. 2f). Mechanically unfolding polyprotein (GB1)$_8$ (eight tandem repeats of GB1) gave rise to sawtooth-like force-extension

curves (Fig. 2g). Each peak (except the last one) corresponds to the unfolding of an individual GB1 domain. The unfolding force is ~180 pN (Fig. 2h), which is significantly higher than the mechanical stability of the TIP-1:Kir complex. This suggests that Gel-1 is very brittle because the LBM did not provide additional extensibility to the gel.

In Gel-2, we used (GB1-HP67)$_4$ domains[50,51] as the LBM (Fig. 2i). Unfolding the hetero-polyprotein (GB1-HP67)$_4$ only gave rise to unfolding force peaks corresponding to GB1 (Fig. 2j, k). Because HP67 alternates with GB1 in the hetero-polyprotein, the long, featureless spacer preceding the GB1 unfolding events in the figure is expected from the mechanical unfolding of HP67, which occurs at forces below the detection limit of our AFM (<10 pN). Therefore, the unfolding of HP67 at minimal forces could potentially provide large extensibility of the gel before rupturing the CLs. Although GB1 was incorporated into the LBM to facilitate the expression of HP67, it did not provide additional extension, due to its high mechanical stability.

Finally, in Gel-3, we used (SH3)$_8$ domains[52] as the LBM (Fig. 2l). The unfolding force of SH3 is ~ 25 pN (Fig. 2m, n), which is sufficiently high to dissipate energy but is lower than the

break force of the CLs (Supplementary Fig. 1b). Therefore, upon stretching Gel-3, the unfolding of SH3 imparted both high extensibility and high toughness to the hydrogel.

The kinetics for CL breakage and the mechanical unfolding of various LBMs are summarized in Supplementary Table 1. Taken together, these results suggest that the mechanical properties of

the CLs and LBMs in the three gels follow distinct mechanical hierarchies, which are expected to lead to different mechanical behaviors of the gels.

**Gels exhibited predicted mechanical behaviors at the bulk level.** Having verified our protein choices for gel design at the single-

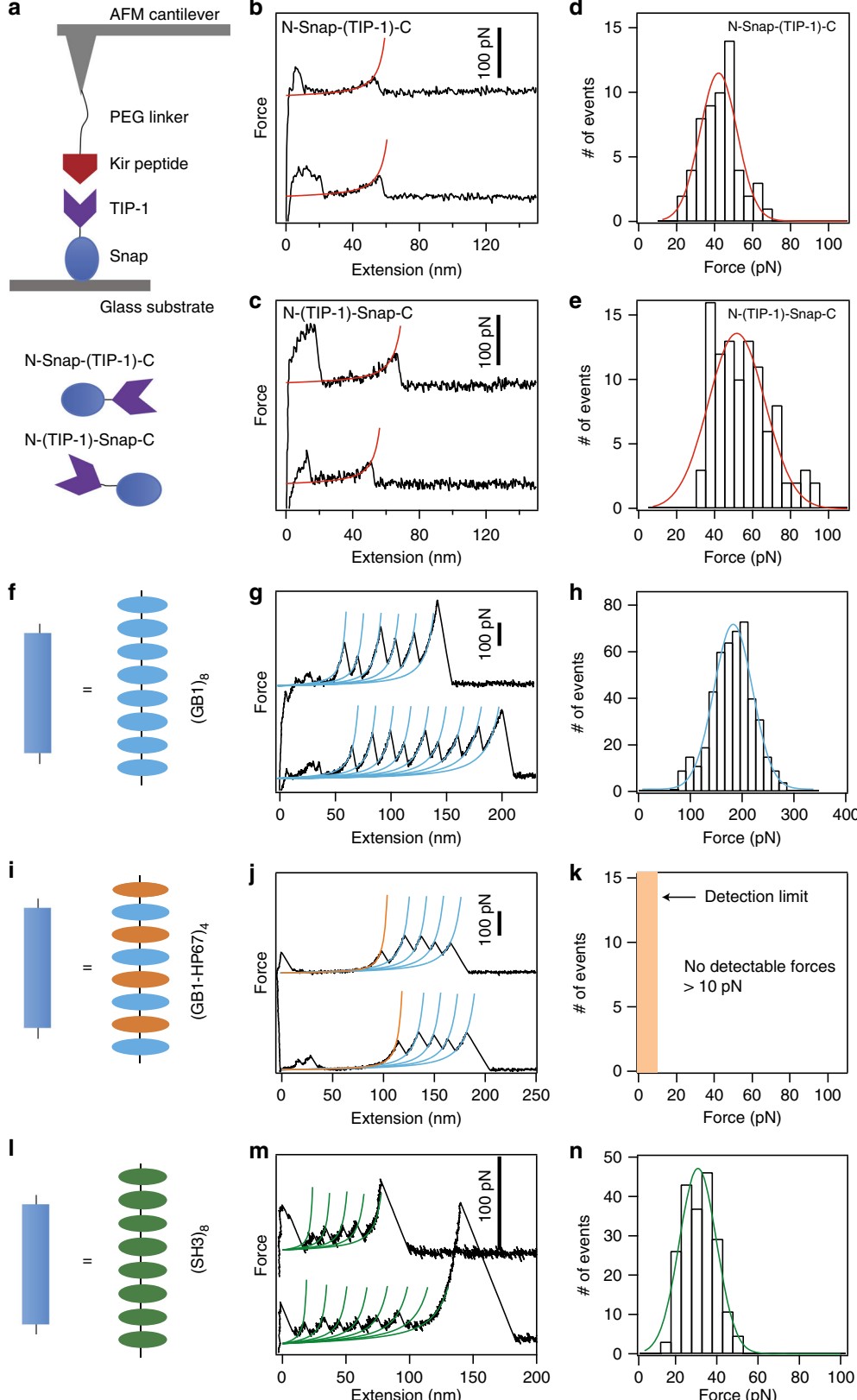

molecule level, we prepared synthetic protein hydrogels for mechanical testing. As depicted in Fig. 3a, a CL aqueous solution consisting of four-armed PEG terminated by Kir peptide on each arm was mixed with an aqueous solution of ABA triblock proteins with TIP-1 at both ends and different LBMs in the center. The molar ratio of Kir and TIP-1 was maintained at 1:1, and the final total ABA protein concentration was 180 mg mL$^{-1}$, unless otherwise specified. When pouring the two solutions into the ring-shape silicone rubber mold, they formed an opaque hydrogel within 2 min, presumably through the formation of TIP-1:Kir complexes. The hydrogels were free-standing and did not swell much in phosphate buffered saline (PBS) (Supplementary Fig. 2). The erosion rate of the hydrogels was also slow (Supplementary Fig. 3).

Then, these gels were subjected to mechanical testing in PBS at room temperature. The typical stress–strain curves of Gel-1, Gel-2, and Gel-3 are shown in Fig. 3c–e. Gel-1 can only extend by ~16% with a Young's modulus of ~150 kPa at a strain of 5%. Gel-2 is highly extensible and can extend by ~250% before failure. The Young's modulus of Gel-2 is 10 times less than that of the Gel-1, but the failure stress is similar in Gel-2. Gel-3 is highly extensible, with a failure strain of ~215% and a Young's modulus of ~27 kPa at a strain of 50%. The higher extensibilities of Gel-2 and Gel-3 compared to Gel-1 may suggest the unfolding of HP67 and SH3 prior to gel fracture. The failure stress of Gel-3 is only slightly higher than those of Gel-1 and Gel-2, which may suggest that the failure stress is strongly affected by the mechanical properties of the cross-linkers (see Supplementary Methods for theoretical analysis details). Decreasing protein concentrations from 180 mg mL$^{-1}$ to 150 mg mL$^{-1}$ leads to decreased Young's modulus and failure stress, but the failure strain remains the same (Supplementary Fig. 4). This may suggest that the force experienced by the protein network at the molecular level at a given strain does not affected by protein concentrations. However, due to the relatively high minimal gelation concentrations (~100 mg mL$^{-1}$) of the hydrogels and the limited solubility of the proteins (~180–190 mg mL$^{-1}$), the effect of protein concentrations on the mechanical properties of hydrogels were only tested in a narrow range of protein concentrations. The mechanical features of the three hydrogels are summarized in Supplementary Tables 2–9 and align perfectly with our projected design specifications.

To further confirm that mainly HP67 and SH3 but not GB1 unfold in the gels on deformation, we subjected the three gels to stretching–relaxation cycles (Fig. 3f–h). The stretching–relaxation cycles for Gel-1 did not exhibit any appreciable hysteresis, further confirming that GB1 does not unfold on deformation. The stretching–relaxation cycles for Gel-2 also did not show any apparent hysteresis, even at a strain of 150%, which may be due to the low unfolding force of HP67 domains[50,51]. By contrast, in Gel-3, we observed clear hysteresis between the stretching and the relaxation curves, even at a low strain of 20% (Fig. 3h). The hysteresis increases at higher strains, indicating more energy dissipated in the stretching–relaxation cycle. Note that because the unfolded protein can quickly refold when force is released, the hydrogel can quickly recover its mechanical properties within a

few seconds, as shown in Supplementary Fig. 5. The initial mechanical properties of Gel-3 are almost completely recovered within 10 s, suggesting that all unfolded SH3 upon stretching in Gel-3 can correctly refold in the stretching–relaxation cycles. Fast and efficient refolding is a unique trait of folded globular proteins[34]. This mechanism is distinct from the high toughness hydrogels using sacrificial bonds for intermolecular cross-linking[53–55], which requires much longer periods (often several hours) and higher temperatures to recover. Moreover, because SH3 can refold under residual forces[52], Gel-3 can partially regain its mechanical properties even if the hydrogel is not completely relaxed to its original length in the stretching–relaxation cycles, but the hysteresis between stretching and relaxation traces becomes less at higher strains. (Supplementary Fig. 6).

The self-healing properties of the three gels were also tested by breaking the hydrogel rings and then put them back to the original molds for different healing time (Supplementary Fig. 7). All three gels (Gels 1–3) can self-heal after damage, indicating that the TIP-1:Kir binding used for hydrogel cross-linking is reversible (Supplementary Fig. 7). The stretching–relaxation cycles of the self-healed hydrogels are similar to those of the pristine ones, other than a reduction in the Young's moduli (Supplementary Figs. 7 and 8), indicating that energy dissipation is mainly controlled by the mechanical properties of the LBMs.

**Mechanics of designed hydrogels correlate with theoretical predictions.** With the use of the experimentally determined mechanical folding/unfolding parameters (Supplementary Table 1), our theoretical predictions confirm that elasticity, extensibility and fracture toughness of the hydrogels are indeed due to the rational design of the proteins at the molecular level (see Supplementary Methods for theoretical analysis details). At the macroscopic level, the cross-linked protein network, schematically shown in Fig. 4a, is modeled using a cubic representative volume element (RVE) (Fig. 4b). At the molecular level, each protein chain behaves as a worm-like chain, with the contour length change corresponding to forced stochastic unfolding/folding of protein domains. The stress–strain curves for Gels 1–3 (Fig. 4c–e) and the corresponding unfolded fractions of folded domains within the gels upon uniaxial loading and unloading cycles are determined and shown in Fig. 4f.

Our theoretical analysis indicates that Gel-1 has a high Young's modulus of 130 kPa but a low extensibility of 17%. Approximately 12.4% GB1 domains are unfolded at the initial free swelling state. Upon loading, the portion of unfolded GB1 remains almost unchanged because the maximum force applied to LBM is limited by the mechanical stability of the CL, which is too weak to accelerate the unfolding and suppress the refolding of GB1 domains. As the LBM of Gel-1 does not provide any additional extensibility, Gel-1 is likely to be very brittle. Gel-2 has a high extensibility of 150% but a low Young's modulus of 15 kPa. HP67 is fully unfolded at the initial free swelling state (100%), which provides great extensibility to Gel-2. Upon loading, a very limited portion of GB1 is additionally unfolded, and therefore GB1 does not provide additional extension as for Gel-1. As the extensibility

**Fig. 2** Single-molecule force spectroscopy of the hydrogel building blocks. **a** Schematic of the SMFS experiments performed to assess the mechanical stability of the TIP-1:Kir complex. Kir peptide was attached to the AFM cantilever through a PEG linker. TIP-1 was anchored on the substrate through a Snap tag. Depending on the protein constructs, TIP-1 can be pulled either from the N-terminus or from the C-terminus. **b, c** Representative force-extension curves for the rupture of the TIP-1:Kir complexes from two different pulling directions. Red lines correspond to worm-like chain (WLC) fittings. **d, e** The rupture force histograms for the two pulling directions. **f** Schematic of the polyprotein (GB1)$_8$. **g** Representative force-extension curves. Each peak represents an unfolding event of GB1 upon stretching. Cyan lines correspond to WLC fittings. **h** The unfolding force histogram. **i** Schematic of the polyprotein of (GB1-HP67)$_4$. **j** Representative force-extension curves. Only the unfolding of GB1 (high force peaks) can be detected in the traces. HP67 is mechanically labile, giving rise to the long featureless region prior to the unfolding events of GB1. **k** HP67 unfolds at forces lower than the detection limit of our AFM (~ 10 pN). **l** Schematic of the polyprotein (SH3)$_8$. **m** Representative force-extension curves. Each peak represents an unfolding event of SH3 upon stretching. Blue lines correspond to WLC fittings. **n** The unfolding force histogram. The pulling speed for all experiments was 400 nm s$^{-1}$

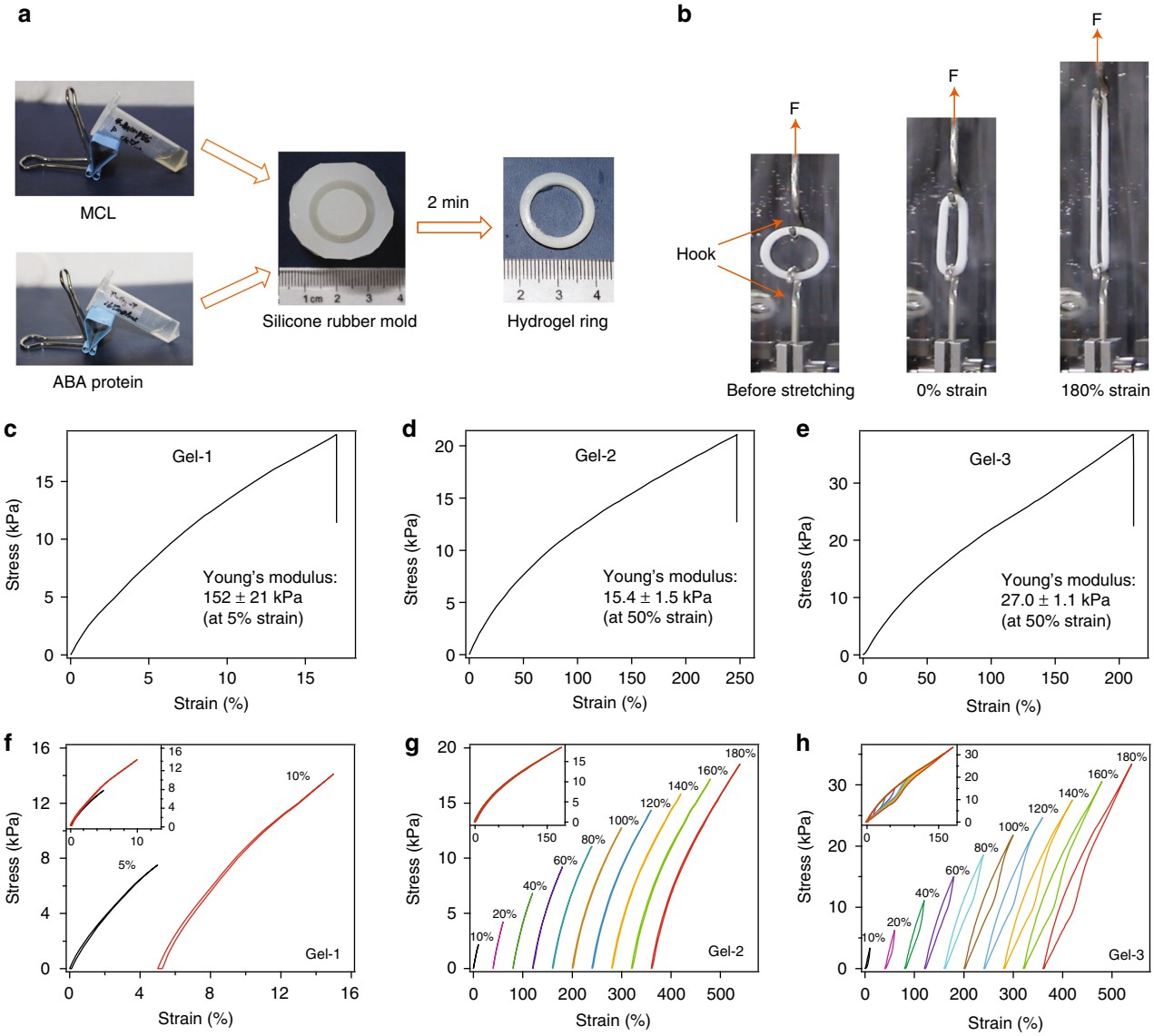

**Fig. 3** The bulk mechanical properties of the designed hydrogels. **a** The general procedure to prepare the protein hydrogel rings for mechanical testing. All protein gels were cross-linked by TIP-1:Kir complexes. (GB1)$_8$, (GB1-HP67)$_4$ and SH3 were used as the LBMs and the corresponding hydrogels were denoted as Gel-1, Gel-2 and Gel-3, respectively. **b** The setup for the mechanical tests of the ring-shaped hydrogel. **c–e** The stress–strain curves for Gel-1, Gel-2 and Gel-3 until break. The protein concentrations were 180 mg mL$^{-1}$. **f–h** Representative stretching–relaxation curves for Gel-1, Gel-2 and Gel-3. The curves are horizontally offset for clarity. The final strains are shown on the curves. Insets show the superposition of the stretching–relaxation curves at different strains

of Gel-2 is mainly provided by the unfolded HP67, little energy is dissipated during loading/unloading cycles. Gel-3 has a moderate initial Young's modulus of 23 kPa and a high extensibility of more than 210%. At the initial free swelling state, 25% of the SH3 domains are unfolded. Upon loading, 30% of the SH3 domains are additionally unfolded, which affords Gel-3 high extensibility. The unfolding of folded SH3 during the unloading process dissipates large amount of energy, which affords high toughness to Gel-3. The relative failure stress for Gel-1, Gel-2, and Gel-3 is calculated to be 1.4:1:2, which is close to the experimental measured value of 1:1:2 shown in Fig. 3c, d. These theoretical predictions are consistent with the experimental data, validating our design principles. Moreover, our theoretical modeling suggests that folded protein domains experience considerable free swelling forces to unfold mechanically weak proteins, consistent with that reported in literature[30].

Note that the failure stress and the shapes of hysteresis for the gels from theoretical calculation are slightly different from that measured in experiments. In our theory, proteins in the gels form perfect networks with mechanical properties being described by RVE shown in Fig. 4b. However, in reality, the cross-linked protein network within fabricated gels might not be so perfect as that illustrated in Fig. 1a. For example, some arms of CLs might not be able to form crosslinks with LBMs and some LBMs might have just hanged to the network from one of its end or both ends might be separated from the network. Due to these defects, the protein length in the main network may be inhomogeneous and unfolding or refolding dynamics of folded domains within LBMs would be affected, which are expected to affect mechanical properties of fabricated gels in turn. In addition, the loading condition was assumed to be uniaxial tension in the simulation, which is slightly different from that in our experiments. These

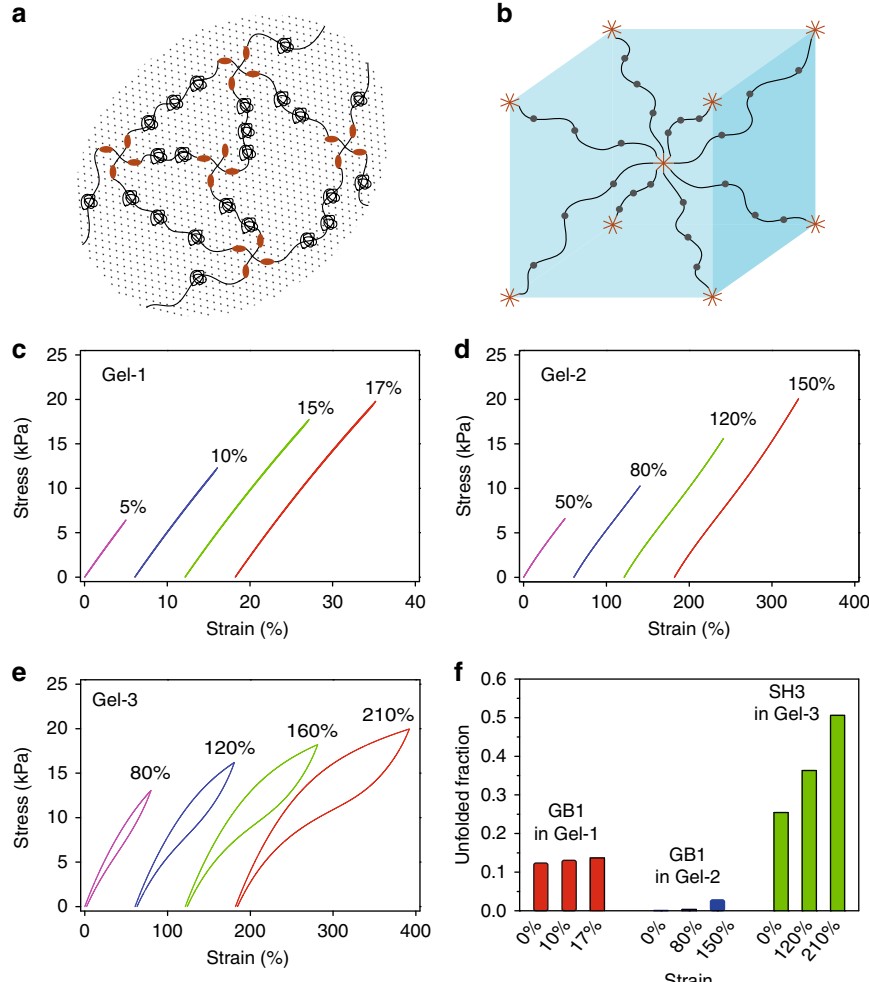

**Fig. 4** Theoretical calculations of the mechanical response of the hydrogels. **a** Schematics of cross-linked protein gels with folded domains; **b** Representative volume element of a cube for gel elasticity. Within this cube, one protein chain extending out from each corner is cross-linked to one arm of the linker proteins extending from the cubic center. **c–e** Prediction of stress–strain curves for Gel-1 (**c**), Gel-2 (**d**), and Gel-3 (**e**) upon uniaxial loading and unloading cycles. Only Gel-3 is shown to dissipate a large amount of energy within loading cycles; (**f**) Prediction of unfolded fractions of folded domains within different gels upon uniaxial loading

might be part of the reasons why the failure stress and the shapes of hysteresis obtained from experiments are different from our theoretical prediction, as comparing Fig. 3 with Fig. 4.

**Rationally designing hydrogels mimics passive mechanics of muscle**. Having established a direct link between the mechanical properties of hydrogels at the macroscopic and the molecular levels, we then set out to rationally engineer a protein hydrogel mimicking the complex passive elasticity of muscle. Muscle is strong, tough, elastic and able to function under significant load and spontaneously self-heal when injured. These properties are essential for the mechanical stability and function of the tissue. The passive mechanical properties of muscle are mainly determined by the elastic properties of titin, which connects the Z-disk and M-line and spans half the length of a sarcomere[56–58]. In previous pioneering work using covalent cross-linking, Li and coworkers elegantly showed that it is possible to mimic the passive elasticity of muscle by engineering titin mimics at the single-molecule level[31]. However, the muscle-mimicking biomaterials were covalently cross-linked and were thus unable to self-heal after breakage. To produce muscle-mimicking materials with self-healing properties, we decided to use strong protein–protein interactions as the cross-linkers. Recently, Nash, Gaub and

coworkers reported a strong protein complex of cohesin (Coh) and X-module-dockerin (Xmod-Doc) from cellulosomes, which can withstand a stretching force higher than the unfolding forces of GB1[59]. If Xmod unfolds before the rupture of the complex, the remaining Doc:Coh dissociates at forces of ~ 300 pN. If Xmod remains folded, the Xmod-Doc:Coh complex dissociates at forces of ~600 pN[59]. However, the Xmod-Doc:Coh complex is only mechanically strong when pulling from the N-terminus of Xmod-Doc and the C-terminus of Coh. We therefore designed an ABA-type triblock protein made of C-terminal disulfide-cross-linked Coh-(GB1)$_4$ and an MCL made of four-armed PEG linked to the N-terminus of Xmod-Doc (Fig. 5a).

Single-molecule AFM experiments confirmed that the GB1 domains can unfold before breakage of Xmod-Doc:Coh (Fig. 5b–d). The unstructured PEG can partially extend and provide entropic elasticity to the gel at low strains, mimicking the unstructured PEVK and N2B sequences in titin[58]. After synthesis of the components, we constructed Gel-4 by mixing the MCL and ABA protein solutions to form an opaque ring (Fig. 5e). Gel-4 had a Young's modulus of ~100 kPa, which is close to that of muscle (Fig. 5f and Supplementary Tables 8 and 9). Moreover, the resulting hydrogel is highly elastic and can be extended to >200% with a failure stress of ~100 kPa (Fig. 5f). Analysis of the

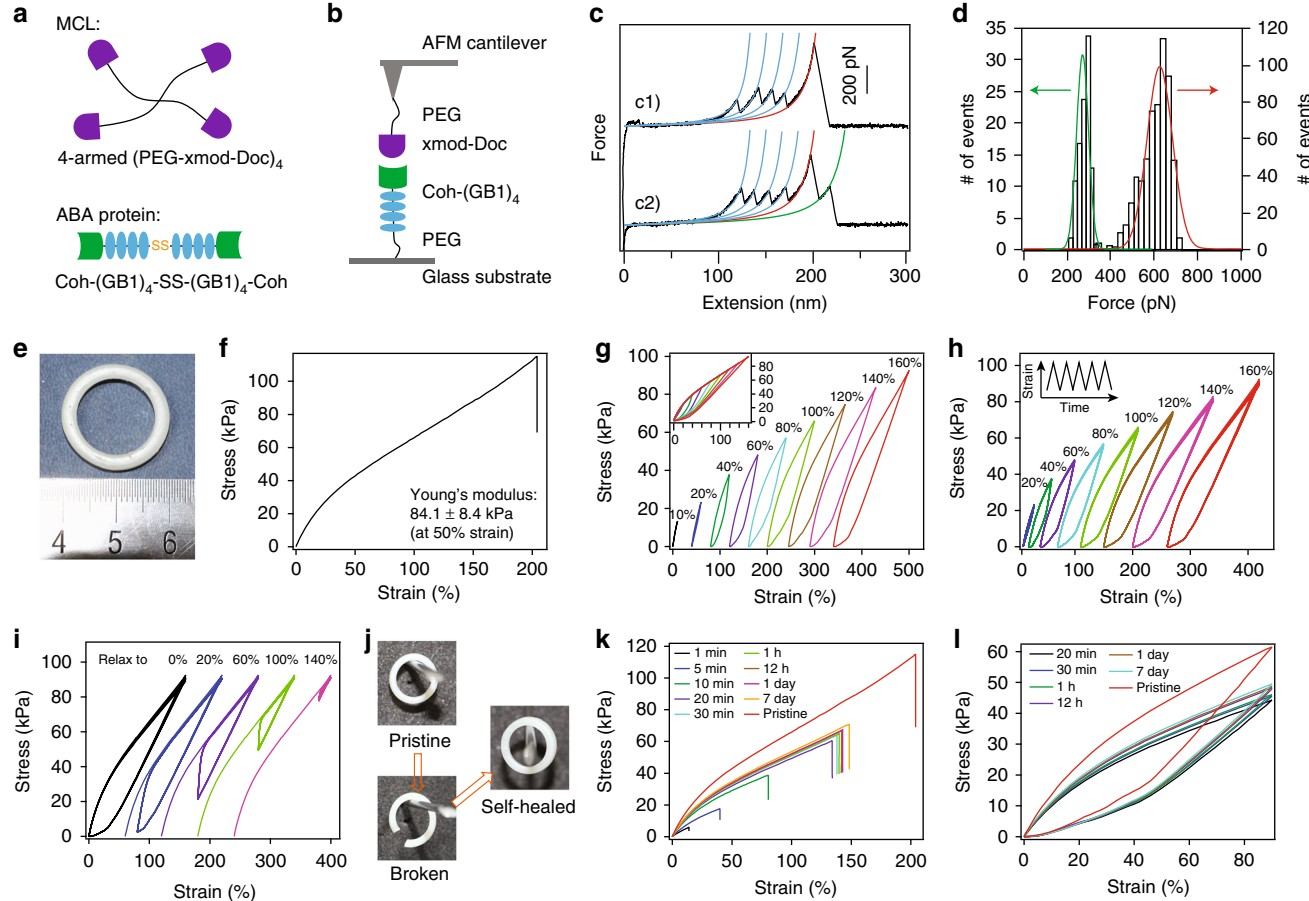

**Fig. 5** Rationally designed self-healable muscle-mimetic hydrogel (Gel-4). **a** Schematic of the building blocks for the hydrogel network. **b** The mechanical properties of the minimum unit in the hydrogel stretched by single-molecule AFM. **c** Two representative force curves. Sawtooth-like patterns with unfolding forces of ~180 pN correspond to the unfolding of GB1 domains. The rupture of the Xmod-Doc:Coh could either result in a single peak with a force ~600 pN (c1, accounts for 40% of total events) or double peaks with forces of ~600 and ~300 pN, respectively (c2, accounts for 60% of total events). **d** The rupture force histograms for the two types of rupture events of the Xmod-Doc:Coh complex. The higher force events for both types of traces are the same and are centered at ~600 pN, which defines the mechanical properties of the Xmod-doc:Coh complex. The lower force histogram for the double-peak events centers at ~300 pN, which is also higher than the unfolding forces of GB1. **e** Photo of the hydrogel ring. **f** The stress–strain curves of the muscle-mimic hydrogel until break. **g** The stretching–relaxation curves of the hydrogel when being stretched to different strains. The curves are horizontally offset for clarity. The final strains are shown on the curves. Insets show the superposition of the stretching–relaxation curves at different strains. **h** Five continuous stretching–relaxation cycles of the hydrogel (Gel-4) to different strains without any waiting time between each cycle. The inset shows the change of strain with time in the cycles. Only minute mechanical fatigue of muscle-mimic hydrogels was observed. **i** The stress–relaxation curves of the hydrogel when first being stretched to a strain of 160% and then immediately relaxed to different strains and stretched back for 5 repetitive cycles. **j** Photos of the pristine, broken and self-healed hydrogels hanging on a peg. **k** The recovery of the mechanical properties of the hydrogel after different self-healing times. **l** The stretching–relaxation curves of self-healed hydrogels with different healing times

stretching–relaxation cycles show that at a low strain of 10%, Gel-4 dissipates almost no energy (Fig. 5g). However, with the increase in the strain to 20% or higher, the stretching and relaxation curves are no longer superimposable. The amount of dissipated energy increases with the increase in strain, indicating that more GB1 domains are unfolded. Because GB1 can refold quickly at a speed of ~700 s$^{-1}$[34], the mechanical properties of the gel are almost completely recovered within the cycle time of our mechanical testing machine (~0.1 s) when relaxing the hydrogel ring to zero extension without waiting (Fig. 5h). More remarkably, the GB1 domains manage to refold even if the hydrogel is only partially relaxed and residual stress remains acting on the hydrogel, due to the extremely high folding ability of GB1 under force (Fig. 5i)[34]. We found that Gel–4 can fully recover its mechanical stress when relaxing to ~100% strain, which is superior to even the mechanical features of covalently cross-linked muscle-mimicking materials[31]. After verifying that

the mechanical properties of Gel-4 reflect those of natural muscle, we tested its self-healing properties. The hydrogel was cut and placed back into the silicone rubber mold for different healing times. As shown in Fig. 5j, its ring shape could fully recover, with no deformation due to the effect of its own weight when suspended from a peg. Its mechanical properties could also partially recover to that of the pristine gel (Fig. 5k). With the increase in healing time to 24 h, the stress at breakage could recover up to 65% of that for the pristine sample. Further prolonging the healing time did not give rise to an appreciable increase in the mechanical properties. We propose that the number of cross-linking bonds that can reform is not limited by the rebinding rate of the specific interaction (less than a second) but controlled by the diffusion of the binding partners on the cut surface. The self-healed gels also showed similar hysteresis to the pristine sample during stretching–relaxation cycles despite the overall stress and strain having decreased, which may indicate

that some of the Xmod-Doc:Coh complexes have not reformed at the breaking interface of the hydrogel (Fig. 5l). Therefore, using a non-covalent protein complex as the cross-linker, it is possible to engineer protein hydrogels with passive mechanical properties similar to those of muscle and that can partially self-heal after breakage.

## Discussion

In nature, different tissues have evolved to possess unique mechanical features for diverse biological functions. For example, articular cartilage is strong (strength of 9–40 MPa), tough (fracture energy of 1,000–15,000 J m$^{-2}$) and elastic (failure strain of 60–120%)[60]; ascending aorta is soft (strength of 0.3–0.8 MPa)[61], whereas mammalian tendon is strong (strength of 50–100 MPa) and resilient (resilience of ~90%) but relatively inextensible (failure strain of 12–16%)[62]. A key challenge in biomaterials research is to produce synthetic hydrogels that can replicate the diverse mechanical properties of the naturally occurring tissues for various biomedical applications. Although previous studies have shown great promise in translating the mechanical properties of polymers and proteins at the single-molecule level to bulk materials, they generally have not considered the important contributions from intermolecular interactions[30,31,63]. A full understanding of the direct link between disparate material scales, from single molecules to the macroscale, remains elusive. Here we demonstrate that the macroscopic mechanical properties of a proteinaceous hydrogel can be predicted by integrating theory, simulation and experimental techniques. On the basis of this, we provide a general principle for the design of protein-based hydrogels that can be specifically tailored with desired mechanical properties. We show that it is possible to independently tune rigidity, toughness, extensibility and self-healing of a protein hydrogel by considering the interplay between the mechanical properties of the crosslinker and the load-bearing modules at the single-molecule level.

Our study provides several important insights into the molecular level determinants of the mechanical properties of protein materials. First, the failure stress of hydrogels is directly correlated with the mechanical stability of the crosslinkers. Because non-covalent interactions are typically mechanically weaker than covalent ones, the upper limit for the failure stress of a protein hydrogel is determined by the covalent cross-linking strength. Second, the elasticity of a hydrogel is mainly determined by the distance between the CL points. As far as the LBMs remain folded in the hydrogels, the hydrogels have similar Young's modulus values, although they may possess distinct mechanical stabilities. However, when LBMs are unfolded in the free swelling state, the Young's modulus of the hydrogel is significantly reduced, similar to that constructed by unstructured proteins. Third, the toughness of a hydrogel is determined by the mechanical stability of both the CLs and the LBM. It is possible to engineer tough protein hydrogels by choosing strong LBMs following a special mechanical hierarchy with CLs. Finally, self-healing properties can be engineered by using specific protein–protein interactions, which can readily break and reform in response to mechanical forces.

Our study also provides strong evidence that individual protein building blocks in a hydrogel can experience forces as high as a few hundred piconewtons at a strain of 200%, which is contradictory to the forces of a few piconewtons predicted in literatures[30]. Such high stretching forces can be achieved only if strong protein crosslinkers are used. Therefore, tough hydrogels can be engineered using mechanically strong folded proteins. If the crosslinkers are strong enough, the toughness of the hydrogel will directly correlate with the mechanical strength of the folded

proteins. Moreover, based on the theoretical modeling, we find that the force in the free swelling hydrogels are around a few piconewtons, which is sufficiently high to shift the folding/ unfolding equilibrium of proteins in the hydrogels. Therefore, protein domains in synthetic hydrogels should be mechanically strong enough to remain folded in order to function properly. These findings suggest that the mechanics of proteins and the mechanical function of their hydrogels are tightly correlated.

With the advances in single-molecule force spectroscopy, the tool box of available proteins and crosslinkers with well-characterized mechanical properties has been significantly expanded in the past several years. It is now possible to rationally tune the mechanical properties of proteins at the single-molecule level[38]. Specifically, many proteins with environmentally responsive mechanical properties have been engineered. We anticipate that these mechanical features can directly translate to mechanical properties in biomaterials.

In addition, many biological materials have hierarchical composite structures[38], suggesting that the design principle illustrated here is not limited to the fabrication of homogeneous protein-based materials. With the knowledge of the mechanical properties of different building blocks such as graphene, carbon nanotubes and self-assembled peptides, it may be also possible to rationally engineer composite biomaterials with hierarchical structures and specifically designed mechanical properties. Combining single-molecule studies with theory and simulations can provide an interesting platform for the design of biomaterials with desired mechanical features for cell culture, tissue engineering and regenerative medicine.

Despite great success in applying this design principle for hydrogel design, caution should be taken when the designed hydrogels involve complex intermolecular interactions. In our hydrogel system, the protein building blocks are generally short. When long and flexible proteins are used, the entanglement of different protein chains may lead to additional physical cross-linking and change the macroscopic mechanical properties. In addition, nonspecific intermolecular interactions in our hydrogel system are also not considered. In typical protein hydrogels, protein concentrations are often very high (>100 mg mL$^{-1}$), close to their solubility limit. Therefore, nonspecific interactions can be significant and may potentially lead to mesoscopic clusters of protein chains and even protein unfolding and aggregation, making the prediction of macroscopic mechanical properties of the hydrogels difficult. Moreover, for cell culture using protein hydrogels, the biochemical properties of different proteins are also important and should be considered seriously. Ideally, the same protein should be used to avoid this issue for the study of the influence of hydrogel mechanics to the cellular behaviors. In order to avoid the use of different proteins, we can use site-directed mutagenesis to tune the mechanical properties of the proteins without changing the chemical properties, which will be our next endeavors. Nonetheless, the design principle illustrated here can also be used to tackle these complicated hydrogel systems, if all kinds of interactions at the molecular level can be fully understood.

## Methods

**Protein engineering**. The gene encoding proteins TIP-1-(GB1)$_8$-TIP-1, TIP-1-(GB1-Hp67)$_4$-TIP-1, TIP-1-(SH3)$_8$-TIP-1, Coh-(GB1)$_4$-cys and cys-Xmod-Doc were constructed in pQE80L vectors using standard molecular biology techniques. The proteins were expressed in *E. coli* (BL21) and purified by Co$^{2+}$-affinity chromatography. The proteins were dialyzed into deionized water and lyophilized before use.

**Single-molecule force spectroscopy**. Single-molecule AFM experiments were carried out on a commercial AFM (ForceRobot 300, JPK, Berlin, Germany) in PBS buffer or TBS buffer at room temperature. The spring constants of the AFM

cantilevers (Biolever-RC-150VB-70 from Olympus or MLCT from Bruker) were calibrated using the equipartition theorem before each experiment, with typical values of 6 and 50 pN nm$^{-1}$, respectively. The pulling speeds were 400 nm s$^{-1}$ for all traces unless otherwise specified.

**Hydrogel preparation and mechanical test**. Hydrogels were prepared in a custom-made silicone rubber mold with a ring-shaped slot ($d_{in}$ = 15 mm, $d_{out}$ = 21 mm, $h$ = 6 mm) based on the specific protein–peptide interaction between TIP-1 and the Kir peptide or the specific protein–protein interaction between Xmod-Doc and Coh. Tensile tests were performed using an Instron-5944 tensometer with a 10 N static load cell at room temperature. Additional characterization of proteins, peptides and the hydrogels can be found in Supplementary Figures 9–14.

**Theoretical modeling**. The mechanical properties of the hydrogels were modeled using the experimentally determined parameters. At the macroscopic level, the cross-linked protein network of a dry synthetic material was represented with a volume element of a cube (RVE). Within this cube, a single-protein chain extending out from each corner is cross-linked to one arm of the linker proteins extending from the cubic center. The schematic of the RVE under uniaxial tension and the simulated local stresses in a ring-shaped hydrogel are shown in Supplementary Figure 15. The parameters used for modeling Gel-1–3 are listed in Supplementary Tables 10–12. The comparison of the simulated results with the experimentally obtained ones are summarized in Supplementary Tables 13–16.

**Data availability**. The data supporting the findings of this study are available from the authors upon reasonable request.

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

## Acknowledgements

This work is funded by the National Natural Science Foundation of China (Nos. 21522402, 11372279, 11572285, 11674153, 11374148, and 11334004), the PAPD of Jiangsu Higher Education, and the Fundamental Research Funds for the Central Universities (020414380070, 020414380058, and 020414380050), and the 973 Program of China (No. 2012CB921801 and 2013CB834100).

## Author contributions

Y.C., B.C. and W.W. conceived the project and designed the experiments. J.W., P.L., C. D., H.J., B.X., X.G. and M.Q. carried out the experiments. J.W., P.L., C.D., H.J., B.X., and X.G. analyzed the data. Y.C, B.C., J.W. and W.W. wrote the manuscript with the input from all authors.

## Additional information

**Competing interests:** The authors declare no competing financial interests.

