## [Peer Review File · Nature Communications]

Reviewers' comments:

Reviewer #1 (Remarks to the Author):

The manuscript "Rationally designed synthetic protein hydrogel with predictable mechanical properties" by Junhua Wu et al. reports a set of experiments and modeling to show that the mechanical properties of hydrogel is predictable based on the building blocks properties and tunable by utilizing different building blocks. This is very interesting and important work.

Especially, their design principle can tune the rigidity, toughness, extensibility, and self-healing of hydrogels, which is shown with the measurement of synthesized hydrogel samples. This reviewer believes that the work is impressive and nicely presented with sufficient results.

However, the mechanical properties predicted from continuum theory are not accurate enough to claim that the properties of the designed hydrogel are fully predictable. This reviewer believes that the authors should share and substantiate more details about their models, analysis for the mismatch, and the possible plans or strategies for the improvement in the future.

Then, this reviewer shall suggest its publication in Nature communication after some minor revisions.

General comments and questions:

1. Summary, line 28. "the nanomechanics of individual building blocks, the correlation has been qualitative rather than quantitative"
-. This reviewer believes the work presented in the manuscript is still qualitative level rather than quantitative. Please tone down or revise the sentence. This sentence may mislead.
2. Figure 2. The scale bar of force is confusing. Please double check j and k. It seems that the force of HP67 cannot reach larger than 10pN in panel k, but panel j shows that several peaks are larger than 100pN.
3. Figure 3. Please explicitly categorize the graphs c-f, d-g, and e-h, indicating Gel-1, 2, and 3.
4. The authors should expand more citations on relevant predictive modeling work, and include e.g. Lin et al, Nat Commun. 2015; 6: 6892, doi: 10.1038/ncomms7892; Rim et al., ACS BSE, doi: 10.1021/acsbmaterials.7b0029

Specific questions on modeling:

1. Figure 2. There is a larger variation of the peak force values and the number of peak force. How this variation incorporates with theoretical modeling.
2. Figure 4. It is difficult to imagine how the simulations are set up. Can the authors include a schematic figure for the geometry of hydrogel in simulation with boundary conditions? Also, it would be great if deformation map with local strain or stress for a hoop sample is included.
3. Figure 4e and Figure 3h. Why the shapes of hysteresis look different from experiment? Please discuss them in the main script.
4. Figure 4b. Please discuss the reliability of the model the author used. It looks BCC structures for crosslinked points. How the trends can be differed with FCC system? The model can capture shear

deformation well?

5. Supplementary tables 10-12. It is hard to capture which parameters are critical to describe the differences. Please provide a summary table for comparison with selected parameters related to rigidity, toughness and extensibility.

6. Please compare the absolute values (toughness, failure strain, Young's modulus) from simulations with those from experiments. Also, please discuss the reasons for the mismatch. It would be great if the authors can discuss how the modeling can be improved.

Reviewer #2 (Remarks to the Author):

Rationally designing hydrogels with predictable and tunable mechanical properties remains a fundamental and enduring challenge in the field of biomaterials. This manuscript from Dr. Cao's lab combines protein engineering, single molecule atomic force microscopy and simulation to address this challenge. They satisfactorily showed that the mechanical properties of the hydrogels at the bulk level are indeed directly correlated with the mechanics at the molecular level. They also provided a general and simple principle, based on the mechanical hierarchy of the crosslinkers and the load-bearing modules, to rationally design the mechanical properties of protein hydrogels. They further demonstrated the successful engineering of hydrogels mimicking the mechanical properties of muscle. This paper also showed that upon stretching the mechanical forces propagated within hydrogels can be as high as a few hundred piconewtons in Gel-4, sufficient to unfold mechanically stable proteins, which is surprising and may change the traditional view on gel mechanics at low strains. Traditionally, it was thought that the force experienced by hydrogel network can only reach high values at extremely high strains and therefore most mechanically stable proteins do NOT unfold in hydrogels at low strains. Their discovery may open great opportunity to use mechanically stable proteins as shock absorbers to increase the toughness of hydrogels.

They also found that the mechanical stability of protein complexes is not directly correlated with their thermodynamic stability and is the deterministic factor for the break strength of hydrogels. This finding is also insightful to scientists working in the field of hydrogel based materials.

Based on the reasons mentioned above, I find this work is solid and novel and is potentially suitable for publications in Nature Communications. However, I have a few comments that should be addressed before its acceptance for publication.

1) In this work the authors used a four-armed PEG as the crosslinker. Could the authors comment on how different crosslinker architecture (e.g. an eight-armed PEG or a linear multivalent polymer) affects the mechanical properties of hydrogels? I am not asking for additional experiments. But, the authors should characterize the four-armed PEG-Kir conjugates to see how many arms of the PEG were indeed linked with the Kir peptide.

2) Can the authors explain why the hydrogels did not swell in PBS buffer? Is it because folded protein domains are more rigid than unstructured polymers? I would suggest the authors quantitatively characterize the volume change and erosion of the hydrogels in PBS buffers.

3) For cell culture using protein hydrogels, the biochemical properties of different proteins are also important and should be considered seriously. Ideally, the same protein should be used to avoid this issue for the study of the influence of hydrogel mechanics to the cellular behaviors. Can the authors comment whether it is possible to engineer protein hydrogels with different mechanical behaviors using the same protein by design?

Minor points:

1) Can the authors explain why there are two types of rupture curves for Xmod-Doc:Coh complexes as shown in Figure 5c?

2) In extended Data Figure 5, c-e, the labels 1m, 5m, 10m, 20m, 30m should be 1 min, 5 min, 10 min, 20 min, 30 min, 1day 7day should be 1 d, 7d.

Reviewer #3 (Remarks to the Author):

The authors present a well-conceived study to design hydrogel systems based on cross-linked elastomeric proteins. The primary claim investigated centers around the relative magnitudes of forces required to break cross-links compared to unfolding/breaking the elastomeric protein. The premise is that by designing specific proteins and crosslink moieties with variable relative strengths, the macroscale properties of hydrogel strength, toughness, extensibility and self-healing can be controlled in a predictable manner. This claim is investigated primarily using single molecule atomic force microscopy at the protein interaction force scale and using mechanical testing and developing a predictive theoretical model at the macroscale hydrogel scale. This principle is then demonstrated for the token application of developing a hydrogel which matches the requirement of skeletal muscle mimics in terms of mechanical properties such as hysteresis regimes and toughness and self-healing recovery from mechanical damage.

The novelty of this principle lies in the demonstration of controlled tunability of hydrogel mechanical properties to match specific biomedical applications, allowing independent controlled experiments evaluating responses to specific changes in macroscale properties rather than treating them as uncontrolled, dependent variables. The broader applications in tissue engineering and rational design of materials make this approach interesting and continue a line of similar articles investigating various combinations of tough hydrogels, elastomeric proteins and protein-crosslink-protein interaction based tailoring of hydrogel properties.

While the work presented is convincing, and the design strategy of both the hydrogels and the study is elegant, some aspects need further clarification.

The authors talk about the effect of decreasing protein concentration and not seeing a difference in failure strain (Line 198-201 and Extended data Figure 2). The range explored seems fairly tight (150 mg/ml vs 180 mg/ml) with little justification for the choice made and the extrapolation of that data for all gel types for various densities seems like an aggressive interpretation. The densities at which hydrogel formation occurs might help provide perspective on the choice of the range.

Line 224-225: "..., but the recovery is slowed down significantly by force." This statement needs to be re-written for clarity. In the same vein, the details on the mechanical testing regimes used for the various experiments are not clear enough for reproduction of data. There is no mention if any pre-conditioning of hydrogel samples were performed for stabilization of viscoelastic properties. Additionally, the relaxation ratios, residual strains and relaxation times picked in the various experiments should be provided in the supplementary section, not just in figures which reduces clarity. In Supplementary materials, Line 266, the load cell ("force gauge") needs to be specified in terms of capacity (peak load) and sensitivity.

Line 264-265: "... Is calculated to be 1.4:1:2, which is close to the experimental measured value of 1:1:2...". This sentence needs further discussion/justification, at the very least suggesting the experimental/theoretical deviations/assumptions respectively which potentially contribute to the discrepancy. This is especially critical since the primary contribution of this work is the claim that a computational model is linked with experimental protein-level characterization to predict the properties of macroscale hydrogels. Thus, simply stating "These theoretical predictions are consistent with the experimental data, validating our design principles." Is not enough.

Line 377-379: "These findings suggest that the mechanics and function of proteins in hydrogels are tightly correlated." This sentence seems to make an unintentionally broad claim. Do the authors intend to state that "These findings suggest that the mechanics of proteins and the mechanical function of their hydrogels are tightly correlated." ? (Because only the mechanical/physical properties of the hydrogels have been investigated in this study).

Readability and attention-to-detail issues:

Line 84-101: There are no references provided for any of the claims in this paragraph. While not expected for the hypotheses that the authors are proposing, some statements especially LBM and CL deformation and fracture mechanisms should be referenced findings. This is specifically requested to appreciate how much of this paragraph is demonstrably known compared to what is a novel proposition.

Line 109: "at" the bulk level, instead of "in"

Line 179: "a" CL instead of "an"

Line 186: close brackets please

Line 283: "breakage" instead of "break", at the authors discretion

Line 370: "contradictory to that predicted in literature." Or alternately "contradicted by". More crucially here, a sentence to explain what the contradiction is and how the authors reconcile it is critical, mere observation of contradiction is unreasonable.

Line 707: "residual" instead of "residule". Also, for consistency, perhaps "offset for clarity" should be mentioned in this figure as well? It is unclear if the loading regime is similar to figure 5(i) and if so, the labeling and chart format should be kept consistent for clarity.

Line Supplementary 280: Symbols introduced should be explicitly introduced (quite a few are assumed to be understood because of how they are used).

Supplementary Table S1. Please specify exactly which data are from literature

Table S4 and S8. Since all tables specify sample size for groups and in these two tables the number of samples per group are different for different conditions, it is recommended to keep table legends consistent across all tables.

No evaluation of statistical power was noted, and no specific statistical comparisons of significance were made. While sample sizes were noted for the studies it would be meaningful to statistically evaluate when there was significant recovery of properties after healing, etc. Which was not immediately noted, but is available from the tables in the supplementary section.

Point-by-point response to reviewers' comments:

Reviewers' comments:

Reviewer #1 (Remarks to the Author):

The manuscript "Rationally designed synthetic protein hydrogel with predictable mechanical properties" by Junhua Wu et al. reports a set of experiments and modeling to show that the mechanical properties of hydrogel is predictable based on the building blocks properties and tunable by utilizing different building blocks. This is very interesting and important work.

Especially, their design principle can tune the rigidity, toughness, extensibility, and self-healing of hydrogels, which is shown with the measurement of synthesized hydrogel samples. This reviewer believes that the work is impressive and nicely presented with sufficient results.

However, the mechanical properties predicted from continuum theory are not accurate enough to claim that the properties of the designed hydrogel are fully predictable. This reviewer believes that the authors should share and substantiate more details about their models, analysis for the mismatch, and the possible plans or strategies for the improvement in the future.

Then, this reviewer shall suggest its publication in Nature communication after some minor revisions.

General comments and questions:

1. Summary, line 28. "the nanomechanics of individual building blocks, the correlation has been qualitative rather than quantitative"- . This reviewer believes the work presented in the manuscript is still qualitative level rather than quantitative. Please tone down or revise the sentence. This sentence may mislead.

Response:

Following this suggestion, we have revised this sentence accordingly.

Revision made:

Main manuscript (Page 2)

“However, it remains challenging to correlate the mechanical properties of protein hydrogels with the nanomechanics of individual building blocks.”

2. *Figure 2. The scale bar of force is confusing. Please double check j and k. It seems that the force of HP67 cannot reach larger than 10pN in panel k, but panel j shows that several peaks are larger than 100pN.*

Response:

We thank the reviewer for this comment. Figure 2j shows the unfolding events for the hetero-polyprotein (GB1-HP67)₄ instead of the homo-polyprotein of HP67. The high force events in the curves correspond to the unfolding of GB1. We have explained this in the previous main text but not the figure legend. We have now also indicated this in the revised legend of Figure 2j and k.

Revision made:

Main manuscript (Page 8 and 30)

“Unfolding the hetero-polyprotein (GB1-HP67)₄ only gave rise to unfolding force peaks corresponding to GB1. Because HP67 alternates with GB1 in the hetero-polyprotein, the long, featureless spacer preceding the GB1 unfolding events in the figure is expected from the mechanical unfolding of HP67, which occurs at forces below the detection limit of our AFM (< 10 pN).”

“j) Representative force-extension curves. Only the unfolding of GB1 (High force peaks) can be detected in the traces. HP67 is mechanically labile, giving rise to the long featureless region prior to the unfolding events of GB1.”

3. *Figure 3. Please explicitly categorize the graphs c-f, d-g, and e-h, indicating Gel-1, 2, and 3.*

Response:

We thank the reviewer for this suggestion and have revised the figure accordingly.

Revision made:

Main manuscript (Page 31)

4. The authors should expand more citations on relevant predictive modeling work, and include e.g. Lin et al, Nat Commun. 2015; 6: 6892, doi: 10.1038/ncomms7892; Rim et al., ACS BSE, doi: 10.1021/acsbmaterials.7b0029

Response:

We thank the reviewer for this suggestion. These references are now cited in the revised manuscript.

Revision Made:

Main manuscript (Page 25)

“41 Lin, S., Ryu, S., Tokareva, O., Gronau, G., Jacobsen, M. M., Huang, W., Rizzo, D. J., Li, D., Staii, C., Pugno, N. M., Wong, J. Y., Kaplan, D. L. & Buehler, M. J. Predictive modelling-based design and experiments for synthesis and spinning of bioinspired silk fibres. *Nature communications* 6, 6892, doi:10.1038/ncomms7892 (2015).

42 Rim, N. G., Roberts, E. G., Ebrahimi, D., Dinjaski, N., Jacobsen, M. M., Martin-Moldes, Z., Buehler, M. J., Kaplan, D. L. & Wong, J. Y. Predicting silk fiber mechanical properties through multiscale simulation and protein design. *ACS Biomater Sci Eng* 3, 1542-1556, doi:10.1021/acsbiomaterials.7b00292 (2017).”

Specific questions on modeling:

1. Figure 2. There is a larger variation of the peak force values and the number of peak force. How this variation incorporates with theoretical modeling.

Response:

The variation of the peak force values is an intrinsic feature of single molecule force spectroscopy experiments because unfolding of folded domains within a polyprotein chain is stochastic. The broadness of the force peak distribution is roughly reversely proportional to the width of the potential well underlying the unfolding pathway. Such broad force distribution can be adequately modeled using the Bell's law. Therefore, in our theory, instead of using the unfolding forces as input parameters directly, we used kinetic parameters (spontaneous unfolding rate and unfolding distance) extracted from the unfolding events based on the Bell's law, as given in Eq. (S5), where the unfolding rate depends on force. Note that the Bell's law is widely used to describe the random breakage of molecular bonds under force and can adequately reproduce the large variation of the peak force values observed in experiments. We then used this unfolding rate together with the folding rate to calculate fraction evolution of unfolded domains within a protein chain upon stretching, as formulated in Eq. (S7).

On the other hand, the number of force peaks are random because we relied on non-specific interactions to pick up the polyproteins from the substrate by AFM cantilever. The anchoring points of the protein to the substrate or the cantilever tip were distributed randomly along the contour of the polyprotein. Only the domains within the anchoring points were stretched. Therefore, number of unfolding events in the force curves was usually less than the total number of domains in the polyprotein. The last peak of the force curves typically corresponds to the detachment of the polyprotein either from the cantilever tip or from the substrate. In our theory, the number of protein domains was fixed as that in the hydrogel design, instead of using the number of force peaks obtained in single molecule experiments. The number of folded protein domains was calculated based on the folding-unfolding kinetics under force. See Eq. (S7) and the text above this equation.

Revision Made:

Supplementary Information (Page 39)

“In single molecule force spectroscopy experiments, the variation of the peak force values is an intrinsic feature because unfolding of folded domains within a polypeptide chain is stochastic. The broadness of the force peak distribution is roughly reversely proportional to the width of the potential well underlying the unfolding pathway. Such broad force distribution can be adequately modeled using the Bell’s law. Therefore, in the theory, we used the kinetic parameters (spontaneous unfolding rate and unfolding distance) extracted from the unfolding events based on the Bell’s law to model unfolding of proteins under force, which can adequately reproduce the large variation of the peak force values observed in experiments.”

2. Figure 4. It is difficult to imagine how the simulations are set up. Can the authors include a schematic figure for the geometry of hydrogel in simulation with boundary conditions? Also, it would be great if deformation map with local strain or stress for a hoop sample is included.

Response:

As shown in Fig. 3b, circular rings made of gels were loaded by two metallic hooks in experiments. Since gels were very soft and these rings were easily to be bended, very small loads (typically ~5 mN) were required to deform rings to a relatively straight configuration in experiments, as shown in the middle of Fig. 3b. This straight configuration was then taken as the initial configuration in our experiments. Beyond this point, gels were considered to be under uniaxial tension, as seen on the right of Fig. 3b. Thus, the stress-stretch curves shown in Figs. (3, 5) were regarded to be obtained under the loading condition of uniaxial tension.

Correspondingly in our theory, we modeled gels with a representative volume element (RVE), which was under uniaxial tension along "1" direction with $\sigma_2 = \sigma_3 = 0$, as illustrated in Supplementary Figure 15.

We have added these simulation details and the schematic figure for the geometry of hydrogel in simulation with boundary conditions, Supplementary Figure 15 a, into the revised Supplemental Information. Strictly speaking, the loading condition in the ring samples was not exactly uniaxial tension in our experiments, as demonstrated by our newly added simulation results of a hoop sample (with a neo-hookean constitutive relation) using ABAQUS (Supplementary Figure 15 b-e). It is expected that the approximation of uni-axial tension to simplify the calculation in our

theory should not affect the theoretical results in any significant ways. It would also be great to improve the theoretical calculation in our future work. We have also commented on this in the revised Supplementary Information.

Revision Made:

Supplementary Information (Page 44 and Page 16)

"As shown in Figure 3b, circular rings made of gels were loaded by two metallic hooks in experiments. Since gels were very soft and these rings were easily to be bended, very small loads were required to deform rings to a straight configuration in experiments, as shown in the middle of Figure 3b. This straight configuration was then taken as the initial configuration in our experiments. Beyond this point, the hydrogels were regarded to be under uni-axial tension, as seen on the right of Figure 3b. Correspondingly in our theory, we modeled gels with a representative volume element under uniaxial tension along "1" direction with $\sigma_2 = \sigma_3 = 0$, as illustrated in Supplementary Figure 15a. However, strictly speaking, the loading condition in the ring samples is not exactly uniaxial tension in our experiments, as suggested by our simulation results of a hoop sample (with a neo-hookean constitutive relation) using ABAQUS (Supplementary Figure 15b-e). It is expected that the approximation of uni-axial tension to simplify the calculation in our theory should not affect the theoretical results in any significant ways. It would also be great to improve the theoretical calculation in our future work. "

Supplementary Figure 15 (a) The RVE is under uniaxial tension along "1" direction with $\sigma_2 = \sigma_3 = 0$ in the theory. (b-e) Simulated deformation and local stresses of a ring-shape hydrogel upon the same loading as that in experiments with ABAQUS. In the simulation, outer diameter of the ring is 20 mm, the inner diameter of the ring is 16 mm, and the cross-section of the ring is circular. The material is chosen to be neo-hookean with an initial Young's modulus of 130 kPa. Due to symmetry, only a quarter of the ring is investigated. The deformed shape and the

contour maps of local stresses are provided: (b) Initial shape and deformed shape of a quarter of a soft ring, (c) Contour map of σ_1 , (d) Contour map of σ_2 , and (e) Contour map of σ_3 .

3. Figure 4e and Figure 3h. Why the shapes of hysteresis look different from experiment? Please discuss them in the main script.

Response:

We thank the reviewer for this comment. In our theory, it was regarded that proteins gels form a crosslinked three-dimensional network with its mechanical properties described by RVE shown in Figure 3b. However, in reality, the crosslinked protein network within fabricated gels might not be so perfect as that illustrated in Figure 1a. For example, some arms of CLs might not be able to form crosslinks with LBMs and some LBMs might have just hanged to the network from one of its end or both ends might be separated from the network. Due to these defects, the protein length in the main network may be inhomogeneous and unfolding or re-folding dynamics of folded domains within LBMs would be affected, which are expected to affect mechanical properties of fabricated gels in turn. In addition, the loading condition was assumed to be uniaxial tension in the theory, which is slightly different from that in our experiments. These might also be part of the reasons why the shapes of hysteresis obtained from experiments look different from our theoretical prediction, as comparing Figure 3 with Figure 4. In response to this comment, we have added a short discussion in the revised form of main manuscript.

Revision Made:

Main manuscript (Page 14)

"Note that the relative failure stress and the shapes of hysteresis for the gels from theoretical calculation are slightly different from that measured in experiments. In our theory, proteins in the gels form perfect networks with mechanical properties being described by RVE shown in Figure 4b. However, in reality, the crosslinked protein network within fabricated gels might not be so perfect as that illustrated in Figure 1a. For example, some arms of CLs might not be able to form crosslinks with LBMs and some LBMs might have just hanged to the network from one of its end or both ends might be separated from the network. Due to these defects, the protein length in the main network may be inhomogeneous and unfolding or re-folding dynamics of folded domains within LBMs would be affected, which are expected to affect mechanical properties of fabricated gels in turn. In addition, the loading condition was assumed to be uniaxial tension in the theory, which is slightly different from that in our experiments. These might be part of the reasons why the relative failure stress and the shapes of hysteresis obtained from experiments are different from our theoretical prediction, as comparing Figure 3 with Figure 4."

4. Figure 4b. Please discuss the reliability of the model the author used. It looks BCC structures for crosslinked points. How the trends can be differed with FCC system? The model can capture shear deformation well?

Response:

Following this comment, we have discussed the reliability of our model in the revised Supplementary Information. The RVE used in our theory was essentially similar to the 8-chain model (Arruda and Boyce Journal of the Mechanics and Physics of Solids 1993 DOI: 10.1016/0022-5096(93)90013-6), which was a classical network model to describe mechanical properties of polymeric materials. The 8-chain model was mathematically simple but was also able to capture various deformation modes, including uniaxial tension, pure shear, and biaxial tension. By now, the 8-chain model has also been incorporated into various theories to describe gel deformation (Wang and Gao Journal of the Mechanics and Physics of Solids 2016 DOI: 10.1016/j.jmps.2016.04.011; Lopez-Menendez and Rodriguez Journal of the Mechanics and Physics of Solids 2016 DOI: 10.1016/j.jmps.2016.01.015). The RVE in 8-chain model may look like a BCC structure for modeling crystals. To the best of our knowledge, there are no specific models for a FCC system of crosslinked polymeric materials. Other polymeric models, such as 3-chain model (Wang and Guth J. Chem. Phys. 1952 DOI: 10.1063/1.1700682), 4-chain model (Flory and Rehner J. Chem. Phys. 1943 DOI: 10.1063/1.1723791; Treloar Trans. Faraday Soc. 1946 DOI: 10.1039/tf9464200083, or full network model (Wu and van der Giessen JMPS 1993 DOI: 10.1016/0022-5096(93)90043-F) may also model our system well and the similar trends can be expected. The 8-chain model we used can capture shear deformation well, as documented in literature. (Boyce and Arruda Rubber Chemistry and Technology 2000 DOI: 10.5254/1.3547602)

Revision made:

Supplemental Information (Page 46)

"The RVE used in our theory was essentially similar to the 8-chain model¹⁷, which was a classical network model to describe mechanical properties of polymeric materials. The 8-chain model¹⁷ was mathematically simple but also able to capture various deformation modes, including uniaxial tension, pure shear, and biaxial tension. By now, the 8-chain model¹⁷ has also been incorporated into various theories to describe gel deformation^{25,26}. Other polymeric models, such as 3-chain model²⁷, 4-chain model^{28,29}, or full network model³⁰ may also model our system well and the similar trends can be expected. The 8-chain model we used can capture shear deformation well, as documented in literature³¹. "

5. *Supplementary tables 10-12. It is hard to capture which parameters are critical to describe the differences. Please provide a summary table for comparison with selected parameters related to rigidity, toughness and extensibility.*

Response:

We thank the reviewer for this comment. In response to this comment, we have added a new table, Table S13, and also the following discussion into the revised form of Supplementary Information.

Revision Made:

Supplementary Information (Page 46-47 and 22)

"Among parameters adopted in our simulation as listed in Tables (S10-S12), ξ , Ω , L_0^{cl} , k^{cl} , γ , λ , μ were the same for Gel-1, Gel-2, and Gel-3. The unfolding rate, the unfolding distance, the refolding rate, or the refolding distance for different domains within LBMs for Gel-1, Gel-2, or Gel-3 were different and taken from experiments, as listed in Supplementary Table 1. The change in contour length when a domain was unfolded was different for different domains and obtained by subtracting respective folded length from respective unfolded length listed in Supplementary Table 1. Other parameters adopted in our simulation, as listed in Supplementary Tables 10-12, are also given in Supplementary Table 13 for better comparison. It is expected that all these parameters would affect simulated rigidity, toughness and extensibility in our theory."

Supplementary Table 13 Comparison of some parameters used in the simulation for three different gels, as also listed in Supplementary Tables (10-12)

	Gel-1	Gel-2	Gel-3
N_1	8	4	8
N_2		4	
L_c^f	20.8nm	17.6nm	7.6nm
N	0.0105 / nm ³	0.0075 / nm ³	0.0103 / nm ³
L_0^{ch}	7nm	6nm	2.7nm

6. *Please compare the absolute values (toughness, failure strain, Young's modulus) from simulations with those from experiments. Also, please discuss the reasons for the mismatch. It would be great if the authors can discuss how the modeling can be improved.*

Response:

In response to this comment, we have added three new tables (Supplementary Tables 14-16) into the Supplemental Information to compare the absolute values from simulations with those from experiments. We have also made discussions on how the modeling could be improved in the Supplementary Information.

Revision Made:

Supplementary Information (Page 47-48 and 21)

" The comparison of Young's modulus, failure strain, and fracture toughness between theory and experiment was provided in Supplementary Tables 14-16, respectively. Note that the fracture toughness from experiments was obtained through Eq. (S26) with measured E and σ_f from experiments and also an assumption of a crack size of $0.5 \mu m$. As seen from Supplementary Tables 14-16, there exist mismatches between experiment and theory.

In our theory, it was regarded that proteins gels form a crosslinked three-dimensional network with its mechanical properties described by RVE shown in Fig. 3b. However, in reality, the crosslinked protein network within fabricated gels might not be so perfect as that illustrated in Figure 1a. For example, some arms of CLs might not be able to form crosslinks with LBMs and some LBMs might have just hanged to the network from one of its end or both ends might be separated from the network. Due to these defects, protein length in the main network may be inhomogeneous and unfolding or re-folding dynamics of folded domains with LBMs would be affected, which are expected to affect mechanical properties of fabricated gels in turn. In addition, the loading condition was assumed to be uniaxial tension in the simulation, which is slightly different from that in our experiments. These might be part of the reasons why the shapes of hysteresis obtained from experiments look different from our theoretical prediction, as comparing Figure 3 with Figure 4.

It should be pointed out that the crack size, denoted as a in Eq. (S26), was not determined. We therefore suggest that the discrepancy of fracture toughness between theory and experiment might be due to the different crack sizes existing within three different gels. Nevertheless, defects in the crosslinked protein network, voids, and other damages, as well as their evolution under loads might also have contributed this discrepancy.

To improve our theory further, inhomogeneity of protein network of fabricated gels may be captured by varying the length, number of domains, or unfolding or re-folding dynamics of domains, etc. of 8 chains within a RVE. To better predict fracture toughness, it is desirable to find out initial defects within gels in experiments. These defects should then be incorporated into a fracture theory, which also considers how these defects evolve. In addition, the breaking and reformation of bonds between LBMs and CLs have not considered in our theory yet, which should be included in predicting the healing process of gels in our future work. It would also be great to simulate the load-displacement curve of hoop samples, as used in the experiments, by

incorporating the constitutive relation developed in our work into a program for finite element method, such as ABAQUS."

Supplementary Table 14 Comparison of Young's modulus between theory and experiments

	Gel-1	Gel-2	Gel-3
Theory	130 kPa	15 kPa	23 kPa
Experiments	152 kPa	15 kPa	27 kPa

Supplementary Table 15 Comparison of failure strain between theory and experiments

	Gel-1	Gel-2	Gel-3
Theory	17%	150%	210%
Experiments	17%	247%	212%

Supplementary Table 16 Comparison of toughness between theory and experiments

	Gel-1	Gel-2	Gel-3
Theory	4.43×10^{-3} N/m	17.6×10^{-3} N/m	46.2×10^{-3} N/m
Experiments	3.78×10^{-3} N/m	48×10^{-3} N/m	83.9×10^{-3} N/m

Reviewer #2 (Remarks to the Author):

Rationally designing hydrogels with predictable and tunable mechanical properties remains a fundamental and enduring challenge in the field of biomaterials. This manuscript from Dr. Cao's lab combines protein engineering, single molecule atomic force microscopy and simulation to address this challenge. They satisfactorily showed that the mechanical properties of the hydrogels at the bulk level are indeed directly correlated with the mechanics at the molecular level. They also provided a general and simple principle, based on the mechanical hierarchy of the crosslinkers and the load-bearing modules, to rationally design the mechanical properties of protein hydrogels. They further demonstrated the successful engineering of hydrogels mimicking the mechanical properties of muscle.

This paper also showed that upon stretching the mechanical forces propagated within hydrogels can be as high as a few hundred piconewtons in Gel-4, sufficient to unfold mechanically stable proteins, which is surprising and may change the traditional view on gel mechanics at low strains. Traditionally, it was thought that the force experienced by hydrogel network can only reach high values at extremely high strains and therefore most mechanically stable proteins do NOT unfold in hydrogels at low strains. Their discovery may open great opportunity to use mechanically stable proteins as shock absorbers to increase the toughness of hydrogels. They also found that the mechanical stability of protein complexes is not directly correlated with their thermodynamic stability and is the deterministic factor for the break strength of hydrogels. This finding is also insightful to scientists working in the field of hydrogel based materials. Based on the reasons mentioned above, I find this work is solid and novel and is potentially suitable for publications in Nature Communications. However, I have a few comments that should be addressed before its acceptance for publication.

1) In this work the authors used a four-armed PEG as the crosslinker. Could the authors comment on how different crosslinker architecture (e.g. an eight-armed PEG or a linear multivalent polymer) affects the mechanical properties of hydrogels? I am not asking for additional experiments. But, the authors should characterize the four-armed PEG-Kir conjugates to see how many arms of the PEG were indeed linked with the Kir peptide.

Response:

We thank the reviewer for this comment. Different crosslinker architectures will certainly change the mechanical properties of the hydrogels, because they change the force distributions on LBMs and CLs. We have added a short discussion in the revised manuscript. We have also characterized the four-armed PEG-Kir conjugates and found that in average 3.9 out of 4 arms of the PEG were successfully linked with Kir peptides.

Revision Made:

Main manuscript (Page 7)

“Note that if the MCL is changed from the four-armed PEG to an eight-armed PEG or a linear multivalent polymer, mechanical properties of the hydrogels are expected to change accordingly, because they change the force distributions on LBMs and CLs within the protein network. However, because LBMs and CLs are connected in series in our experimental design, they are always subjected to the same loading force irrespective to the architectures of the MCLs. As such, it is expected that the mechanical trends for different hydrogels can be similar with different MCLs.”

Supplementary Information (Page 32 and Page 13)

“The conjugation ratio increased with the increase of the molar ratios of the peptide to maleimide group (Supplementary Figure 12b) and in average ~3.9 out of 4 arms of the PEG were successfully linked with Kir peptides as determined by ITC titration (Supplementary Figure 12c).”

Supplementary Figure 12 Characterization of MCL of 4-armed PEG-Kir. a) HPLC trace of the MCL indicates the high purity of the purified MCL sample. b) The conjugation ratios of 4-armed PEG-maleimide after reacting with different molar ratios of kir peptides as determined by UV-absorbance at 280 nm during dialysis. The conjugation ratio was ~ 3.9 at the experimental conditions used for preparing MCL. c) The conjugation ratio of the MCL was further determined using ITC by titrating the protein solution of TIP-1-(GB1)₄-TIP-1 with 4-armed PEG-Kir. The conjugation ratio of Kir to 4-armed PEG was estimated to be ~3.90 (=2/0.513), which was consistent with that determined by UV-Vis absorbance. The dissociation constant was estimated to be ~1.21 μM.

2) Can the authors explain why the hydrogels did not swell in PBS buffer? Is it because folded protein domains are more rigid than unstructured polymers? I would suggest the authors quantitatively characterize the volume change and erosion of the hydrogels in PBS buffers.

Response:

The hydrogels made of folded protein domains typically do not swell too much in PBS buffer. It might be because they are relatively hard to change their conformations comparing with the unstructured proteins due to the existence of special intra-chain interactions. Following the reviewer's suggestion, we have characterized the volume change and the erosion of the hydrogels in PBS buffers. The new data are included in the revised Supplementary Information.

Revision Made:

Supplementary Information (Page 33-34 and 2-3)

“The hydrogels made of folded protein domains typically do not swell too much in PBS buffer. It might be because they are relatively hard to change their conformations comparing with the unstructured proteins due to the existence of special intra-chain interactions.”

Supplementary Figure 2 Swelling ratios of gels (180 mg mL^{-1}) in PBS buffer at room temperature. The ring-shaped hydrogels were weighted immediately after being taken out of the molds, and the weight was recorded as W_0 . Then the hydrogels were soaked in PBS, pH 7.4, at the room temperature. After certain time, the hydrogel rings were taken out of PBS buffer, blotted onto tissue paper to remove excess buffer and weighted as W_t . The swelling ratio was

calculated according to the formula: Swelling ratio (%) = $(W_t - W_0) / W_0 \times 100\%$. Two different samples were measured and the average value was reported. The error bars represent standard deviation.

Supplementary Figure 3 Erosion profiles of 100 mg of Gel 1-4 (at 180 mg mL^{-1}) with a surface area of 0.86 cm^2 at room temperature in 100 mM phosphate buffer, pH 7.4. The erosion was too slow to be reliably determined using UV spectra if the gel samples were kept still in solution. Therefore, the erosion was measured under mild mechanical shaking. In the experiment, 100 mg of hydrogel was transferred into a cylindrical glass tube with a flat bottom (1.05 cm diameter). The glass tube with the hydrogel was then centrifuged at 1700 g for 10 minutes to completely flat down hydrogel sample to the bottom and smooth the surface of the hydrogel. The hydrogel was allowed to stand overnight. The thin gel film together with the glass tube was then soaked in 5 mL of 100 mM phosphate buffer, pH 7.4, in a scintillation vial. The whole setup was placed on a compact rocker tilting at 50 rpm with amplitude of $\pm 9^\circ$, at room temperature. The erosion profiles were determined by measuring the protein absorbance at 280 nm of the supernatant at successive time points using Nano-drop ultraviolet-visible spectrophotometer. Two different samples were measured and the average value was reported. Error bars represent standard deviation of the experimental data. a) Erosion profile of Gel-1. A linear regression (solid line) measures an erosion rate of $3.55 \times 10^{-4} \text{ mg cm}^{-2} \text{ min}^{-1}$. b) Erosion profile of Gel-2. A linear regression (solid line) measures an erosion rate of $4.54 \times 10^{-4} \text{ mg cm}^{-2} \text{ min}^{-1}$. c) Erosion profile

of Gel-3. A linear regression (solid line) measures an erosion rate of $4.76 \times 10^{-4} \text{ mg cm}^{-2} \text{ min}^{-1}$. d) Erosion profile of Gel-4. A linear regression (solid line) measures an erosion rate of $2.79 \times 10^{-4} \text{ mg cm}^{-2} \text{ min}^{-1}$.

3) *For cell culture using protein hydrogels, the biochemical properties of different proteins are also important and should be considered seriously. Ideally, the same protein should be used to avoid this issue for the study of the influence of hydrogel mechanics to the cellular behaviors. Can the authors comment whether it is possible to engineer protein hydrogels with different mechanical behaviors using the same protein by design?*

Response:

We thank the reviewer for this comment. We totally agree with the reviewer that the biochemical properties of the hydrogels are critical for biomedical applications. In order to avoid the use of different proteins, we can use site-directed mutagenesis to tune the mechanical properties of the proteins without changing the chemical properties. We have now commented this in the revised manuscript.

Revision Made:

Main manuscript (Page 21)

“Moreover, for cell culture using protein hydrogels, the biochemical properties of different proteins are also important and should be considered seriously. Ideally, the same protein should be used to avoid this issue for the study of the influence of hydrogel mechanics to the cellular behaviors. In order to avoid the use of different proteins, we can use site-directed mutagenesis to tune the mechanical properties of the proteins without changing the chemical properties, which will be our next endeavors.”

Minor points:

1) *Can the authors explain why there are two types of rupture curves for Xmod-Doc:Coh complexes as shown in Figure 5c?*

Response:

The rupture of Xmod-Doc:Coh can proceed in two different pathways. If Xmod unfolds before the rupture of the complex, the remaining Doc:Coh dissociates at forces of $\sim 300 \text{ pN}$. If Xmod remains folded, the Xmod-Doc:Coh complex dissociates at forces of $\sim 600 \text{ pN}$. We have commented on this in the revised manuscript.

Revision Made:

Main manuscript (Page 15)

“If Xmod unfolds before the rupture of the complex, the remaining Doc:Coh dissociates at forces of ~ 300 pN. If Xmod remains folded, the Xmod-Doc:Coh complex dissociates at forces of ~ 600 pN.”

2) In extended Data Figure 5, c-e, the labels 1m, 5m, 10m, 20m, 30m should be 1 min, 5 min, 10 min, 20 min, 30 min, 1day 7day should be 1 d, 7d.

Response:

We thank for this comment. We have corrected this in the revised manuscript.

Revision Made:

Supplementary Information (Page 8)

Reviewer #3 (Remarks to the Author):

The authors present a well-conceived study to design hydrogel systems based on crosslinked elastomeric proteins. The primary claim investigated centers around the relative magnitudes of forces required to break crosslinks compared to unfolding/breaking the elastomeric protein. The premise is that by designing specific proteins and crosslink moieties with variable relative strengths, the macroscale properties of hydrogel strength, toughness, extensibility and self-healing can be controlled in a predictable manner. This claim is investigated primarily using single molecule atomic force microscopy at the protein interaction force scale and using mechanical testing and developing a predictive theoretical model at the macroscale hydrogel scale. This principle is then demonstrated for the token application of developing a hydrogel which matches the requirement of skeletal muscle mimics in terms of mechanical properties such as hysteresis regimes and toughness and self-healing recovery from mechanical damage.

The novelty of this principle lies in the demonstration of controlled tunability of hydrogel mechanical properties to match specific biomedical applications, allowing independent controlled experiments evaluating responses to specific changes in macroscale properties rather than treating them as uncontrolled, dependent variables. The broader applications in tissue engineering and rational design of materials make this approach interesting and continue a line of similar articles investigating various combinations of tough hydrogels, elastomeric proteins and protein-crosslink-protein interaction based tailoring of hydrogel properties.

While the work presented is convincing, and the design strategy of both the hydrogels and the study is elegant, some aspects need further clarification.

The authors talk about the effect of decreasing protein concentration and not seeing a difference in failure strain (Line 198-201 and Extended data Figure 2). The range explored seems fairly tight (150 mg/ml vs 180 mg/ml) with little justification for the choice made and the extrapolation of that data for all gel types for various densities seems like an aggressive interpretation. The densities at which hydrogel formation occurs might help provide perspective on the choice of the range.

Response:

We thank the reviewer for this comment. We agree with the reviewer that the range of protein concentrations we used is tight. In our system, the minimal gelation concentrations were determined to be ~ 100 mg/ml by the vial inversion method. But when the concentrations are below 140 mg/ml, the ring-shaped hydrogels are not strong enough to be taken out from the mold. On the other hand, the maximum solubility of the proteins is ~180-190 mg/ml. Therefore,

we have only a very narrow range to study the effect of protein concentrations on the mechanical properties. We have commented this and toned down our initial claim in the revised manuscript.

Revision Made:

Main manuscript (Page 10-11)

“However, due to the relatively high minimal gelation concentrations ($\sim 100 \text{ mg mL}^{-1}$) of the hydrogels and the limited solubility of the proteins ($\sim 180\text{-}190 \text{ mg mL}^{-1}$), the effect of protein concentrations on the mechanical properties of hydrogels were only tested in a narrow range of protein concentrations.”

Line 224-225: “.., but the recovery is slowed down significantly by force.” This statement needs to be re-written for clarity. In the same vein, the details on the mechanical testing regimes used for the various experiments are not clear enough for reproduction of data. There is no mention if any pre-conditioning of hydrogel samples were performed for stabilization of viscoelastic properties. Additionally, the relaxation ratios, residual strains and relaxation times picked in the various experiments should be provided in the supplementary section, not just in figures which reduces clarity. In Supplementary materials, Line 266, the load cell (“force gauge”) needs to be specified in terms of capacity (peak load) and sensitivity.

Response:

We thank the reviewer for these comments and suggestions. We have changed “but the recovery is slowed down significantly by force.” to “but the hysteresis between stretching and relaxation traces becomes less at higher strains.” in the revised manuscript. We have also included the details on the mechanical testing regimes used for the various experiments in the revised Supplementary Information.

Revision Made:

Main manuscript (Page 12, Line 2)

“but the hysteresis between stretching and relaxation traces becomes less at higher strains.”

Supplementary Information (Page 34-37)

“The force gauge (load cell) used for all experiments is a 10 N Static Load Cell (Instron, Catalog no. 2530-10N; Force capacity: $\pm 10 \text{ N}$; Linearity: $\pm 0.25\%$ of reading from 0.4 to 100% of force capacity; Repeatability: 0.25% of reading from 0.4 to 100% of force capacity; Temperature Effect on Sensitivity: $\pm 0.002\%$ of force capacity per $^{\circ}\text{C}$ (0.001% per $^{\circ}\text{F}$).”

“The hydrogel was pre-equilibrated in the buffer for at least 1 h before mechanical testing. No additional pre-conditioning of the hydrogel sample was performed for stabilization of viscoelastic properties.”

“For stretching-relaxation experiments, the hydrogels were first gently stretched to flatten the ring and align it to the force direction. This position was set as zero strain. The hydrogels were stretched from this position to a given strain (provided in the figure) and then relaxed to zero strain or a specific residual strain (provided in the figure) at a strain rate of 20 mm min^{-1} .”

“The test regimes for all the experiments are list below:

Figure 3

In figure 3c-e, the hydrogel was stretched until break at a constant strain rate of 20 mm min^{-1} .

In figure 3f, the hydrogel was stretched to the given length (5% or 10% strain, respectively), and then immediately relaxed to 0% strain. All the experiments were at a constant strain rate of 20 mm min^{-1} .

In figure 3g-h, the hydrogel was stretched to the given length (10%, 20%, 40%, 60%, 80%, 100%, 120%, 140%, 160% or 180% strain, respectively), and then immediately relaxed to 0% strain. All the experiments were at a constant strain rate of 20 mm min^{-1} .

Figure 5

In figure 5f, the hydrogel (Gel-4) was stretched until break at a constant strain rate of 20 mm min^{-1} .

In figure 5g, the hydrogel (Gel-4) was stretched to the given length (10%, 20%, 40%, 60%, 80%, 100%, 120%, 140%, 160% strain, respectively), and then immediately relaxed to 0% strain. All the experiments were at a constant strain rate of 20 mm min^{-1} .

In figure 5h, the hydrogel (Gel-4) was stretched to the given length (20%, 40%, 60%, 80%, 100%, 120%, 140%, 160% strain, respectively), and then immediately relaxed to 0% strain. There are totally 5 continuous cycles without any waiting time between each cycle. All the experiments were at a constant strain rate of 20 mm min^{-1} .

In figure 5i, the hydrogel (Gel-4) was stretched to 160% strain. Immediately after that, the hydrogel (Gel-4) was relaxed to the given residual strain (0%, 20%, 60%, 100%, 140% strain, respectively). There are totally 5 continuous cycles without any waiting time between each cycle. All the experiments were at a constant strain rate of 20 mm min^{-1} .

In figure 5k, the pristine or the self-healed hydrogel (Gel-4) was stretched until break at a constant strain rate of 20 mm min^{-1} .

In figure 5l, the pristine or the self-healed hydrogel (Gel-4) was stretched to 160% strain, and then immediately relaxed to 0% strain. All the experiments were at a constant strain rate of 20 mm min^{-1} .

Supplementary Figure 2

In Supplementary Figure 2, the hydrogel was stretched until break at a constant strain rate of 20 mm min^{-1} .

Supplementary Figure 3

In Supplementary Figure 3, the hydrogel (Gel-3) was stretched to the given length (20%, 40%, 60%, 100%, 140%, 160%, 180% strain, respectively), and then immediately relaxed to 0% strain. There are totally 5 continuous cycles with indicated waiting time between each cycle. All the experiments were at a constant strain rate of 20 mm min⁻¹.

Supplementary Figure 4

In Supplementary Figure 4, the hydrogel (Gel-2 or Gel-3) was stretched to 180% strain. Immediately after that, the hydrogel was relaxed to the given residual strain. There are totally 5 continuous cycles without any waiting time between each cycle. All the experiments were at a constant strain rate of 20 mm min⁻¹.

Supplementary Figure 5

In Supplementary Figure 5c-e, the pristine or the self-healed hydrogel was stretched until break at a constant strain rate of 20 mm min⁻¹.

Supplementary Figure 6

In Supplementary Figure 6, the pristine or the self-healed hydrogel was stretched to 140% strain, and then immediately relaxed to 0% strain. All the experiments were at a constant strain rate of 20 mm min⁻¹.”

Line 264-265: “.... Is calculated to be 1.4:1:2, which is close to the experimental measured value of 1:1:2...”. This sentence needs further discussion/justification, at the very least suggesting the experimental/theoretical deviations/assumptions respectively which potentially contribute to the discrepancy. This is especially critical since the primary contribution of this work is the claim that a computational model is linked with experimental protein-level characterization to predict the properties of macroscale hydrogels. Thus, simply stating “These theoretical predictions are consistent with the experimental data, validating our design principles.” Is not enough.

Response:

In our theory, we first calculated $\bar{\sigma}$, which was found to be 92.3 zJ for Gel-1, 464 zJ for Gel-2, and 977 zJ for Gel-3, respectively. By assuming that the crack size was the same for three different gels, the ratio among failure stresses for Gel-1, Gel-2, and Gel-3 was calculated to be 1.4:1:2 with Eqs. (S26-S27), which was close to the experimental value, 1:1:2, shown in Figs. 3c-d. It should be pointed out that the crack size, denoted by a in Eq. (S26), was estimated to be on the order of 0.5 μm in our calculation, although it is not experimentally determined yet. We then suggest that the discrepancy between theory and experiment might be due to the different crack sizes existing within three gels. Nevertheless, defects in the crosslinked protein network,

voids, and other damages might also have contributed this discrepancy. In response to this comment, we have added following discussion into the revised manuscript and Supplementary Information.

Revision Made:

Supplementary Information (Page 48)

"It should be pointed out that the crack size, denoted as a in Eq. (S26), was not experimentally determined. We suggest that the discrepancy of fracture toughness between theory and experiment might be due to the different crack sizes existing within three different gels. Nevertheless, defects in the crosslinked protein network, voids, and other damages, as well as their evolution under loads might have also contributed this discrepancy."

Main manuscript (Page 14)

"Note that the relative failure stress and the shapes of hysteresis for the gels from theoretical calculation are slightly different from that measured in experiments. In our theory, proteins in the gels form perfect networks with mechanical properties being described by RVE shown in Figure 4b. However, in reality, the crosslinked protein network within fabricated gels might not be so perfect as that illustrated in Figure 1a. For example, some arms of CLs might not be able to form crosslinks with LBMs and some LBMs might have just hanged to the network from one of its end or both ends might be separated from the network. Due to these defects, protein length in the main network may be inhomogeneous and unfolding or re-folding dynamics of folded domains within LBMs would be affected, which are expected to affect mechanical properties of fabricated gels in turn. In addition, the loading condition was assumed to be uniaxial tension in the simulation, which is slightly different from that in our experiments. These might be part of reasons why the relative failure stress and the shapes of hysteresis obtained from experiments are different from our theoretical prediction, as comparing Figure 3 with Figure 4."

Line 377-379: "These findings suggest that the mechanics and function of proteins in hydrogels are tightly correlated." This sentence seems to make an unintentionally broad claim. Do the authors intend to state that "These findings suggest that the mechanics of proteins and the mechanical function of their hydrogels are tightly correlated." ? (Because only the mechanical/physical properties of the hydrogels have been investigated in this study).

Response: We thank the reviewer for this suggestion. We have revised the manuscript accordingly.

Revision Made:

Main manuscript (Page 20)

“These findings suggest that the mechanics of proteins and the mechanical function of their hydrogels are tightly correlated.”

Readability and attention-to-detail issues:

Line 84-101: There are no references provided for any of the claims in this paragraph. While not expected for the hypotheses that the authors are proposing, some statements especially LBM and CL deformation and fracture mechanisms should be referenced findings. This is specifically requested to appreciate how much of this paragraph is demonstrably known compared to what is a novel proposition.

Response:

Following this suggestion, we have cited three references to support the statement of LBM and CL deformation and fracture mechanisms.

Revision Made:

Main manuscript (Page 5 and 25)

“44 Mouw, J. K., Ou, G. & Weaver, V. M. Extracellular matrix assembly: a multiscale deconstruction. *Nature reviews. Molecular cell biology* **15**, 771-785, doi:10.1038/nrm3902 (2014).

45 Wang, H. & Heilshorn, S. C. Adaptable hydrogel networks with reversible linkages for tissue engineering. *Advanced materials* **27**, 3717-3736, doi:10.1002/adma.201501558 (2015).

46 Li, H. & Cao, Y. Protein mechanics: from single molecules to functional biomaterials. *Acc Chem Res*, doi:10.1021/ar100057a (2010).”

Line 109: “at” the bulk level, instead of “in”

Response:

We thank the reviewer for this comment. This typo has been corrected in the revised manuscript.

Revision Made:

Main manuscript (Page 6)

“at the bulk level”

Line 179: “a” CL instead of “an”

Response:

We thank the reviewer for this comment. This typo has been corrected in the revised manuscript.

Revision Made:
Main manuscript (Page 9)

“a CL”

Line 186: close brackets please

Response:
We thank the reviewer for this comment. This typo has been corrected in the revised manuscript.

Revision Made:
Main manuscript (Page 10)

“(PBS)”

Line 283: “breakage” instead of “break”, at the authors discretion

Response:
We thank the reviewer for this comment. This typo has been corrected in the revised manuscript.

Revision Made:
Main manuscript (Page 16)
“breakage”

*Line 370: “contradictory to that predicted in literature.” Or alternately “contradicted by”.
More crucially here, a sentence to explain what the contradiction is and how the authors
reconcile it is critical, mere observation of contradiction is unreasonable.*

Response:
We thank the reviewer for this comment. We have corrected the typo and added a sentence
explain what the contradiction is and how we reconcile it in the revised manuscript.

Revision Made:
Main manuscript (Page 19)

“which is contradictory to the forces of a few piconewtons predicted in literatures. Such high
stretching forces can be achieved only if strong protein crosslinkers are used.”

Line 707: “residual” instead of “residule”. Also, for consistency, perhaps “offset for clarity” should be mentioned in this figure as well? It is unclear if the loading regime is similar to figure 5(i) and if so, the labeling and chart format should be kept consistent for clarity.

Response:

We thank the reviewer for this comment. We have corrected the typos accordingly and mentioned “offset for clarity” in the figure legend (See Supplementary Information, Page 7). The loading regime for this Figure is the same as Figure 5(i) (See Supplementary Information, Page 36). Therefore, we have modified this figure to keep the labeling and chart format consistent with Figure 5(i).

Revision Made:

Supplementary Information (Page 7 and 36)

“**Supplementary Figure 6** The stretching-relaxation cycles of Gel-2 and Gel-3 at different residual strain. a) The experimental protocol. The hydrogel (Gel-2 or Gel-3) was stretched to 180% strain. Immediately after that, the hydrogel was relaxed to the given residual strain. There are totally 5 continuous cycles without any waiting time between each cycle. All the experiments were at a constant strain rate of 20 mm min⁻¹. b) The stress-strain curves for Gel-2. c) The stress-strain curves for Gel-3. The curves in b) and c) are offset for clarity.”

“Supplementary Figure 6

In Supplementary Figure 6, the hydrogel (Gel-2 or Gel-3) was stretched to 180% strain. Immediately after that, the hydrogel was relaxed to the given residual strain following the experimental protocol shown in Supplementary Figure 6a. There are totally 5 continuous cycles

without any waiting time between each cycle. All the experiments were at a constant strain rate of 20 mm min⁻¹.”

Line Supplementary 280: Symbols introduced should be explicitly introduced (quite a few are assumed to be understood because of how they are used).

Response:

We thank the reviewer for this comment. We have explicitly introduced all symbols in the revised manuscript.

Revision Made:

Supplementary Information (Page 37-46)

The modifications are highlighted in blue. Because this part is too long, we did not copy the revised text to the Point-to-point response.

Supplementary Table S1. Please specify exactly which data are from literature

Response:

We have specified this in the revised Supplementary Information.

Revision Made:

Supplementary Information (Page 17)

“The unfolding and folding kinetics for GB1, unfolding and folding kinetics for HP67, folding kinetics for SH3, and unfolding kinetics for Coh:Xmod-Doc were taken from Ref. 1, 2, 3, and 4, respectively.¹⁻⁴”

Table S4 and S8. Since all tables specify sample size for groups and in these two tables the number of samples per group are different for different conditions, it is recommended to keep table legends consistent across all tables.

Response:

We have made the table legends consistent in the revised manuscript.

Revision Made:

Supplementary Information (Page 17-20)

Supplementary Table 2 Mechanical properties of ring-shaped Gel-1 after different gelation time in the mold

Gelation time	Number of samples	Failure strain (%)	Failure stress (kPa)	Failure modulus (kPa)	Young's modulus at 5% (kPa)	Young's modulus at 10% (kPa)
2 min	6	16.72±2.11	17.72±3.29	112.67±19.58	152.31±21.13	133.46 ±27.48
1 d	6	16.88±1.96 ^{ns}	17.54±3.03 ^{ns}	107.23±15.49 ^{ns}	157.22±18.76 ^{ns}	127.91 ±21.17 ^{ns}
7 d	6	16.57±2.03 ^{ns,NS}	17.84±2.91 ^{ns,NS}	110.37±17.18 ^{ns,NS}	149.05±20.44 ^{ns,NS}	128.16 ±19.35 ^{ns,NS}

The hydrogel was stretched until break (failure) at a constant strain rate of 20 mm min⁻¹. The local slope at a given strain on the stress-strain curve was taken as the Young's modulus at this strain. Student's t-test was used for statistical analyses. When p value is below 0.05, it is considered being statistically significant. Compared with 2 min group: **ns**, not significant; Compared with 1 d group: **NS**, not significant.

Supplementary Table 3 Mechanical properties of ring-shaped Gel-2 after different gelation time in the mold

Gelation time	Number of samples	Failure strain (%)	Failure stress (kPa)	Failure modulus (kPa)	Young's modulus at 50% (kPa)	Young's modulus at 100% (kPa)	Young's modulus at 150% (kPa)	Young's modulus at 200% (kPa)
2 min	6	247.63±27.15	21.68±2.39	8.51±0.93	15.44±1.51	11.60±1.18	10.32±1.05	9.27±0.96
1 d	6	245.26±22.17 ^{ns}	22.11±2.23 ^{ns}	8.79±0.81 ^{ns}	15.18±1.41 ^{ns}	11.71±1.11 ^{ns}	10.12±1.18 ^{ns}	9.17±1.03 ^{ns}
7 d	6	249.12±21.39 ^{ns,NS}	20.49±2.56 ^{ns,NS}	8.21±0.85 ^{ns,NS}	15.36±1.47 ^{ns,NS}	11.38±1.03 ^{ns,NS}	10.46±1.14 ^{ns,NS}	9.29±0.92 ^{ns,NS}

The hydrogel was stretched until break (failure) at a constant strain rate of 20 mm min⁻¹. The local slope at a given strain on the stress-strain curve was taken as the Young's modulus at this strain. Student's t-test was used for statistical analyses. When p value is below 0.05, it is considered being statistically significant. Compared with 2 min group: **ns**, not significant; Compared with 1 d group: **NS**, not significant.

Supplementary Table 4 Mechanical properties of ring-shaped Gel-3 after different gelation time in the mold

Gelation time	Number of samples	Failure strain (%)	Failure stress (kPa)	Failure modulus (kPa)	Young's modulus at 50% (kPa)	Young's modulus at 100% (kPa)	Young's modulus at 150% (kPa)
2 min	3	212.18±18.89	38.03±3.21	17.69±1.22	27.04±1.08	22.11±1.17	19.36±1.20
1 d	9	211.23±22.17 ^{ns}	38.45±3.11 ^{ns}	17.98±1.12 ^{ns}	27.74±1.36 ^{ns}	22.09±1.31 ^{ns}	19.44±1.27 ^{ns}
7 d	3	209.47±21.36 ^{ns,NS}	38.25±3.16 ^{ns,NS}	18.23±1.07 ^{ns,NS}	27.63±1.21 ^{ns,NS}	22.32±1.23 ^{ns,NS}	19.47±1.15 ^{ns,NS}

The hydrogel was stretched until break (failure) at a constant strain rate of 20 mm min⁻¹. The local slope at a given strain on the stress-strain curve was taken as the Young's modulus at this strain. Student's t-test was used

for statistical analyses. When p value is below 0.05, it is considered being statistically significant. Compared with 2 min group: **ns**, not significant; Compared with 1 d group: **NS**, not significant.

Supplementary Table 5 Mechanical properties of ring-shaped Gel-1 at different concentration

Protein Concentration (mg/mL)	Number of samples	Failure strain (%)	Failure stress (kPa)	Failure modulus (kPa)	Young's modulus at 5% (kPa)	Young's modulus at 10% (kPa)
150	9	16.42±1.81	15.08±2.37	98.75±12.34	149.63±16.31	121.71 ±19.54
180	9	16.88±1.96 ^{ns}	17.54±3.03 ^{ns}	107.23±15.49 ^{ns}	157.22±18.76 ^{ns}	127.91 ±21.17 ^{ns}

The hydrogel was stretched until break (failure) at a constant strain rate of 20 mm min⁻¹. The local slope at a given strain on the stress-strain curve was taken as the Young's modulus at this strain. Student's t-test was used for statistical analyses. When p value is below 0.05, it is considered being statistically significant. **ns**, not significant.

Supplementary Table 6 Mechanical properties of ring-shaped Gel-2 at different concentration

Protein Concentration (mg/mL)	Number of samples	Failure strain (%)	Failure stress (kPa)	Failure modulus (kPa)	Young's modulus at 50% (kPa)	Young's modulus at 100% (kPa)	Young's modulus at 150% (kPa)	Young's modulus at 200% (kPa)
150	9	243.35±26.18	19.05±2.03	7.83±0.61	13.93±1.31	10.89±1.07	9.41±0.86	8.38±0.81
180	9	245.26±22.17 ^{ns}	22.11±2.23 ^{**}	8.79±0.81 [*]	15.18±1.41 ^{ns}	11.71±1.11 ^{ns}	10.12±1.18 ^{ns}	9.17±1.03 ^{ns}

The hydrogel was stretched until break (failure) at a constant strain rate of 20 mm min⁻¹. The local slope at a given strain on the stress-strain curve was taken as the Young's modulus at this strain. Student's t-test was used for statistical analyses. When p value is below 0.05, it is considered being statistically significant. **ns**, not significant, ^{*}*P* < 0.05, ^{**}*P* < 0.01.

Supplementary Table 7 Mechanical properties of ring-shaped Gel-3 at different concentration

Protein Concentration (mg/mL)	Number of samples	Failure strain (%)	Failure stress (kPa)	Failure modulus (kPa)	Young's modulus at 50% (kPa)	Young's modulus at 100% (kPa)	Young's modulus at 150% (kPa)
150	9	207.49±21.59	32.02±2.74	15.42±0.82	26.83±1.59	18.51±1.17	16.38±1.12
180	9	211.23±22.17 ^{ns}	38.45±3.11 ^{***}	17.98±1.12 ^{***}	22.74±1.36 ^{***}	22.09±1.31 ^{***}	19.44±1.27 ^{***}

The hydrogel was stretched until break (failure) at a constant strain rate of 20 mm min⁻¹. The local slope at a given strain on the stress-strain curve was taken as the Young's modulus at this strain. Student's t-test was used for statistical analyses. When p value is below 0.05, it is considered being statistically significant. **ns**, not significant, ****P* < 0.001.

Supplementary Table 8 Mechanical properties of ring-shaped Gel-4 after different gelation time in the mold

Gelation time	Number of samples	Failure strain (%)	Failure stress (kPa)	Failure modulus (kPa)	Young's modulus at 50% (kPa)	Young's modulus at 100% (kPa)	Young's modulus at 150% (kPa)
2 min	3	203.34±20.15	106.07±10.83	53.18±5.68	84.12±8.42	65.12±6.85	58.13±6.89
1 d	9	203.41±19.72 ^{ns}	108.26±9.64 ^{ns}	53.92±5.31 ^{ns}	85.56±7.93 ^{ns}	65.87±6.72 ^{ns}	58.62±6.61 ^{ns}
7 d	3	203.78±19.14 ^{ns, NS}	108.79±9.22 ^{ns, NS}	53.38±5.02 ^{ns, NS}	86.33±7.12 ^{ns, NS}	66.11±6.34 ^{ns, NS}	58.79±5.82 ^{ns, NS}

The hydrogel was stretched until break (failure) at a constant strain rate of 20 mm min⁻¹. The local slope at a given strain on the stress-strain curve was taken as the Young's modulus at this strain. Student's t-test was used for statistical analyses. When p value is below 0.05, it is considered being statistically significant. Compared with 2 min group: **ns**, not significant; Compared with 1 d group: **NS**, not significant.

Supplementary Table 9 Mechanical properties of ring-shaped Gel-4 at different concentration

Protein Concentration (mg/mL)	Number of samples	Failure strain (%)	Failure stress (kPa)	Breaking modulus (kPa)	Young's modulus at 50% (kPa)	Young's modulus at 100% (kPa)	Young's modulus at 150% (kPa)
150	9	197.26±17.33	95.15±9.37	48.21±4.12	72.24±7.13	55.93±6.12	49.79±4.47
180	9	203.41±19.72 ^{ns}	108.26±9.64 ^{**}	53.92±5.31 [*]	85.56±7.93 ^{**}	65.87±6.72 ^{**}	58.62±6.61 ^{**}

The hydrogel was stretched until break (failure) at a constant strain rate of 20 mm min⁻¹. The local slope at a given strain on the stress-strain curve was taken as the Young's modulus at this strain. Student's t-test was used for statistical analyses. When p value is below 0.05, it is considered being statistically significant. **ns**, not significant, **P* < 0.05, ***P* < 0.01.

No evaluation of statistical power was noted, and no specific statistical comparisons of significance were made. While sample sizes were noted for the studies it would be meaningful to statistically evaluate when there was significant recovery of properties after healing, etc. Which was not immediately noted, but is available from the tables in the supplementary section.

Response:

We have made statistical comparisons of significance and noted evaluation of statistical power in Supplementary Tables 2-9 in the revised Supplementary Information. We also statistically evaluated the recovery of properties after healing for different time. The time point after which there was significant recovery of properties was indicated in Supplementary Figure 13.

Revision Made:

Supplementary Information (Page 48 and 14-15)

“Statistical analysis

Igor Pro and Microsoft Excel was used to plot and analyze all the graphs. Student’s t test was used for statistical analyses. When p value is below 0.05, it is considered being statistically significant.”

Supplementary Figure 13 Failure strains of the pristine and the various time self-healed hydrogels. The hydrogel was stretched until break (failure) at a constant strain rate of 20 mm min⁻¹. Student’s t-test was used for statistical analysis. When p value is below 0.05, it is considered being statistically significant. NS, not significant, $*P < 0.05$. The time point after which the recovery of the mechanical properties is not significant ($P < 0.05$) is defined as the recovery time. Based on this definition, the recovery time for all gels was ~ 20 min.

Supplementary Figure 14 Failure stress of the pristine and the various time self-healed hydrogels. The hydrogel was stretched until break (failure) at a constant strain rate of 20 mm min⁻¹. Student's t-test was used for statistical analyses. When p value is below 0.05, it is considered being statistically significant. NS, not significant, **P* < 0.05. The time point after which the recovery of the mechanical properties is not significant (*P* < 0.05) is defined as the recovery time. Based on this definition, the recovery time for all gels was ~ 20 min.

REVIEWERS' COMMENTS:

Reviewer #1 (Remarks to the Author):

Well done. I recommend the paper for publication.

Reviewer #2 (Remarks to the Author):

The authors have addressed all comments and I suggest the acceptance of the paper as is.

Reviewer #3 (Remarks to the Author):

All the comments and concerns raised in the review were satisfactorily addressed. The revisions made to the manuscript by the authors are thorough and improve the overall manuscript.